# An hourglass circuit motif transforms a motor program via subcellularly localized muscle calcium signaling and contraction

Steven R Sando[1], Nikhil Bhatla[1,2,3], Eugene LQ Lee[1,3], H Robert Horvitz[1]*

[1]Howard Hughes Medical Institute, Department of Biology, McGovern Institute for Brain Research, Massachusetts Institute of Technology, Cambridge, United States; [2]Miller Institute, Helen Wills Neuroscience Institute, Department of Molecular and Cellular Biology, University of California, Berkeley, Berkeley, United States; [3]Department of Brain and Cognitive Sciences, Massachusetts Institute of Technology, Cambridge, United States

**Abstract** Neural control of muscle function is fundamental to animal behavior. Many muscles can generate multiple distinct behaviors. Nonetheless, individual muscle cells are generally regarded as the smallest units of motor control. We report that muscle cells can alter behavior by contracting subcellularly. We previously discovered that noxious tastes reverse the net flow of particles through the *C. elegans* pharynx, a neuromuscular pump, resulting in spitting. We now show that spitting results from the subcellular contraction of the anterior region of the pm3 muscle cell. Subcellularly localized calcium increases accompany this contraction. Spitting is controlled by an 'hourglass' circuit motif: parallel neural pathways converge onto a single motor neuron that differentially controls multiple muscles and the critical subcellular muscle compartment. We conclude that subcellular muscle units enable modulatory motor control and propose that subcellular muscle contraction is a fundamental mechanism by which neurons can reshape behavior.

*For correspondence:
horvitz@mit.edu

Competing interests: The authors declare that no competing interests exist.

## Introduction

How animal nervous systems differentially control muscle contractions to generate the variety of flexible, context-appropriate behaviors necessary for survival and reproduction is a fundamental problem in neuroscience. In many cases, distinct behaviors must be performed using the same muscle or set of muscles. For example, the muscles of the human jaw and tongue control both feeding and speech, while the *Drosophila* wing muscles produce both flight and song (*O'Sullivan et al., 2018*). Such modulation of muscle function can be achieved via neuronally mediated alteration of the amplitudes and relative timing of muscle motions (*Briggman and Kristan, 2008*; *Marder and Calabrese, 1996*).

We are analyzing the neuromuscular control of behavior using the *C. elegans* feeding organ, the pharynx, as a model neuromuscular system. The pharynx (*Figure 1A*) is a neuromuscular pump that contracts rhythmically ('pumps') to generate the suction that ingests bacterial food (*Avery and You, 2012*). The pharynx contains 14 classes of neurons (20 neurons in total), 8 classes of muscles (20 muscle cells in total), 4 glands, and 16 structural cells, and the connectome of these cells has been completely described (*Albertson and Thomson, 1976*; *Cook et al., 2020*).

Behavioral functions are known for 9 of the 14 pharyngeal neuron classes (15 of the 20 total neurons). The two M2 neurons, single M4, and two MC neurons promote pumping and/or ingestion (*Avery and Horvitz, 1987*; *Avery and Horvitz, 1989*; *Trojanowski et al., 2014*); the two M3 and single I5 neurons regulate the timing of muscle relaxation (*Avery, 1993*); the two I2 neurons inhibit pumping (*Bhatla and Horvitz, 2015*); the two I1 neurons promote or inhibit pumping dependent on

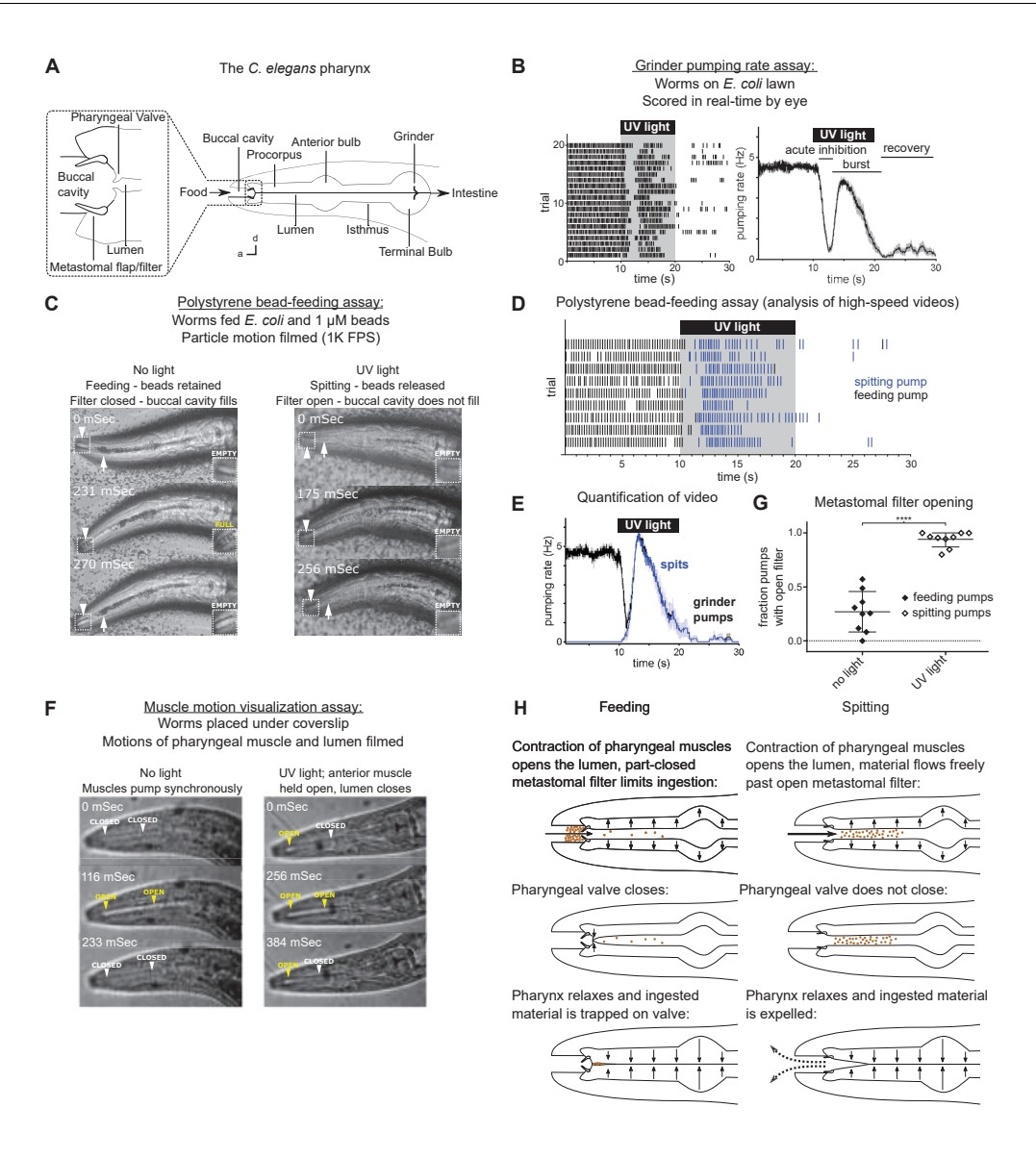

**Figure 1.** Worms spit as a consequence of the sustained contraction of anterior pharyngeal muscles. (A) Anatomy of the *C. elegans* pharynx; a, anterior; d, dorsal. (B) Grinder pumping response to 365 nm UV light. Left, raster plot of individual animals, each grinder contraction shown by a tick; right, backward moving average of the pumping rate of the same animals. Labels indicate the 'acute inhibition,' 'burst,' and 'recovery' phases of the response. Phases previously described by *Bhatla and Horvitz, 2015*. Scale bar, 20 μm. (C) Representative images of worms feeding and spitting in the polystyrene bead feeding assay. Pumps were scored as 'feeding' if they resulted in the retention of particles in the anterior pharynx (arrows), while pumps were scored as 'spitting' if they released beads from the procorpus into the environment or if beads ingested during a procorpus contraction were not retained. In feeding pumps, the metastomal filter was typically partially closed, such that food accumulated in the buccal cavity (arrowheads and inset). Spitting pumps involved the opening of the metastomal filter such that the buccal cavity did not fill. (D) Raster plot of representative results from polystyrene bead feeding assay. Animals carried the transgene *nIs678 (M1_{promoter}::gcamp6s)*. 'M1_{promoter}' refers to the *glr-2* promoter. (E) Quantification of average grinder pumping and spitting rates of animals shown in (D). (F) Representative images of the pumping cycles of feeding (left column) and spitting worms (right column). When open, the lumen of the pharynx (arrowheads) is visible as a white line. The food-trapping pharyngeal valve (left arrowhead) is located at the anterior end of the lumen. In the absence of UV light (left column), the lumen and pharyngeal valve open (yellow arrowheads) and close (white arrowheads) nearly synchronously, enabling the trapping and eventual ingestion of food (*Fang-Yen et al., 2009*). By contrast, in the presence of UV light (right column) the opening of the lumen and the pharyngeal valve become uncoupled. While the lumen continues to open and close, the muscle region at the site of the pharyngeal valve undergoes sustained contraction and thus opens the valve such that the conclusion of a pump ejects material from the pharynx. Animals were *unc-29(e1072)* mutants to facilitate video capture. (G) Quantification of frequencies at which the metastomal filter was open in feeding and spitting pumps in (D). Center bar, mean; error bars, standard deviation (SD). ****, p < 0.0001; t test. (H) Diagrammatic summary of feeding and spitting behavior. Shading around traces indicates standard error of the mean (SEM).

*Figure 1 continued on next page*

*Figure 1 continued*

The online version of this article includes the following source data and figure supplement(s) for figure 1:

**Source data 1.** Source data for *Figure 1*.
**Figure supplement 1.** The odor of hydrogen peroxide induces spitting.
**Figure supplement 1—source data 1.** Source data for *Figure 1—figure supplement 1*.

context (*Dent et al., 2000*; *Trojanowski et al., 2014*; *Bhatla et al., 2015*); the two NSM neurons regulate locomotion and feeding in response to food or food-related odors (*Sawin et al., 2000*; *Li et al., 2012*; *Flavell et al., 2013*; *Rhoades et al., 2019*); and, as described in more detail below, the single M1 neuron controls spitting behavior (*Bhatla et al., 2015*).

We previously found that short-wavelength violet or ultraviolet (UV) light inhibits *C. elegans* feeding behavior and that this modulation likely is mediated by light-generated reactive oxygen species (ROS) and thus reflects a response to noxious taste stimuli (*Bhatla and Horvitz, 2015*). Consistent with this hypothesis, the same set of genes and neurons necessary for light-induced feeding inhibition is required for the inhibition of feeding by the ROS hydrogen peroxide ($H_2O_2$) (*Bhatla and Horvitz, 2015*). Because we and others have found that light and ROS act similarly to elicit behavioral responses (*Hill and Schaefer, 2009*; *Kim et al., 2013b*; *Kim and Johnson, 2014*; *Guntur et al., 2015*; *Du et al., 2016*; *Arenas et al., 2017*; *Birkholz and Beane, 2017*; *Guntur et al., 2017*), and because of the high precision with which light can be controlled as an experimental stimulus, we are using light as a tool to analyze how the neuromusculature of the pharynx produces behavioral responses to noxious stimuli.

We found that upon exposure to short-wavelength light, pharyngeal pumping rapidly stops ('acute inhibition') and then transiently resumes ('burst pumping'; e.g., see *Figure 1B*). After the removal of light, the pumping rate is depressed and slowly recovers ('recovery') to the 4–5 Hz of normal feeding over about a minute (*Bhatla and Horvitz, 2015*). Our previous studies implicated four classes of pharyngeal neurons in the pumping response to light (*Bhatla and Horvitz, 2015*; *Bhatla et al., 2015*). Light inhibits pumping by altering the activity of the I1, I2, and MC neurons. Subsequently, the M1 motor neuron promotes the burst pumping phase, during which the net flow of particles through the anterior pharynx is reversed, producing a spitting behavior that ejects ingested material from the pharynx.

We now report that light transforms feeding pumps into spitting pumps—that is reverses the direction in which particles are driven by pharyngeal pumping—by inducing sustained contractions of subcellular regions of individual muscle cells, the three pm3 pharyngeal muscle cells. Subcellularly localized calcium signals occur in those regions of the pm3 muscles undergoing contraction. Both spitting and the associated localized muscle calcium response are controlled by the M1 neuron. M1 sits at the waist of an 'hourglass' circuit motif, in which multiple upstream sensory neurons converge onto M1, which in turn makes divergent outputs onto multiple muscle classes that control distinct subcomponents of the spitting reflex. Weak activation of M1 via a subset of upstream neurons can evoke a subset of M1's motor outputs, producing an attenuated variant of spitting and showing how graded neuronal activation can produce context-appropriate variations of a core behavior. Together, our results provide an example of how muscle function can be modulated at a subcellular level to produce flexible, contextual behavior and identify subcellularly localized calcium as a mechanistic basis of such modulation.

## Results

### Worms spit as a consequence of the sustained contraction of anterior pharyngeal muscles

We previously defined burst pumping as the transient increase in pumping rate that follows acute light-induced pumping inhibition (*Figure 1B*; *Bhatla and Horvitz, 2015*). Because 365 nm (UV) light evokes burst pumping most robustly (*Bhatla and Horvitz, 2015*), we used this wavelength in the experiments described below. We further characterized the burst pumping phase and observed that there are three alterations to pharyngeal function that occur during burst pumping: (a) as described previously (*Bhatla and Horvitz, 2015*; *Bhatla et al., 2015*), pumping rate increases; (b) the valve at

the anterior end of the pharynx that closes to capture food during feeding instead remains open, causing pharyngeal pumps to conclude with the spitting of pharyngeal contents; and (c) a filter that otherwise restricts particle influx opens, facilitating the rinsing of the pharyngeal lumen (*Figure 1A*).

In this work, we first confirmed that pumping rate increases during burst pumping, as reported previously (*Bhatla and Horvitz, 2015*). We scored pumping in the burst pumping phase in real time by eye, using the assay validated by *Bhatla and Horvitz, 2015* and based on the rate of contraction of the grinder ('grinder pumps'), a readily observed cuticular structure in the terminal bulb (*Figure 1A and B*).

We next confirmed and extended the previous observation that worms spit during the burst pumping phase (*Bhatla et al., 2015*). We distinguished feeding and spitting pumps by recording high-speed videos of worms feeding on a mixture of *E. coli* bacterial food and comparably-sized 1 μm polystyrene beads (*Fang-Yen et al., 2009*; *Bhatla et al., 2015*). We designated pumps that trapped ingested material in the anterior pharynx as feeding pumps (*Video 1*) and pumps that failed to trap material as spitting pumps (*Video 2*; *Figure 1C*). We previously found that violet light (436 nm) induces spitting (*Bhatla et al., 2015*) and have replicated this result using UV light (365 nm) (*Figure 1C–E*). Consistent with the hypothesis that the pumping response to light reflects a response to light-generated ROS, the odor of $H_2O_2$ also induced spitting (*Figure 1—figure supplement 1A*). While light reversed the net anterior-to-posterior flow of beads that occurs during feeding through the anterior pharynx, light did not reverse this anterior-to-posterior movement of beads in more posterior pharyngeal regions, that is, the anterior bulb, isthmus, and posterior bulb (*Figure 1A*).

We identified the biomechanical mechanism by which net particle flow is reversed to produce spitting. In feeding, contraction of the pharyngeal muscles opens the lumen of the pharynx, generating suction and drawing food and fluid inwards. Two cuticular structures project from pharyngeal muscle into the lumen of the anterior pharynx and buccal cavity: the metastomal flaps and the pharyngeal valve (*Figure 1A*; *Albertson and Thomson, 1976*). The metastomal flaps were previously shown to limit the size and flow of particles that enter the pharyngeal lumen from the buccal cavity (*Fang-Yen et al., 2009*). For this reason, we will refer to the metastomal flaps as the 'metastomal filter.' The pharyngeal valve is located immediately posterior to the filter and prevents the escape of ingested particles from the pharyngeal lumen to the buccal cavity: prior to the onset of pharyngeal muscle relaxation, the pharyngeal valve closes, sealing the anterior end of the lumen to retain bacteria while fluid is released via channels that border the lumen (*Fang-Yen et al., 2009*). Given the importance of the pharyngeal valve in feeding, we hypothesized that spitting involves a modulation of the motions of this structure. We recorded videos of animals feeding and spitting under a coverslip such that the anterior pharyngeal muscles and lumen were clearly visualized. As reported previously (*Fang-Yen et al., 2009*), feeding pumps involved the closure of the pharyngeal valve, thus trapping food (*Figure 1F*; *Video 3*). By contrast, spitting pumps were characterized by a strikingly

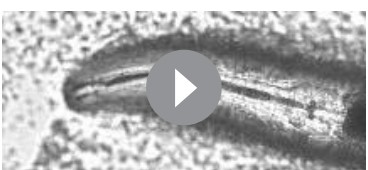

**Video 1.** Feeding pumps in polystyrene bead feeding assay. High frame rate video of *C. elegans* feeding in polystyrene bead feeding assay. The small, dark particles are polystyrene beads. During typical feeding pumps, beads accumulate in the buccal cavity. Beads filter gradually from the buccal cavity into the pharyngeal lumen, where they are trapped by the closure of the pharyngeal valve at the front of the pharynx upon the conclusion of each pump. Video was recorded at 1000 frames per second. Playback is at 3% of original speed.

https://elifesciences.org/articles/59341#video1

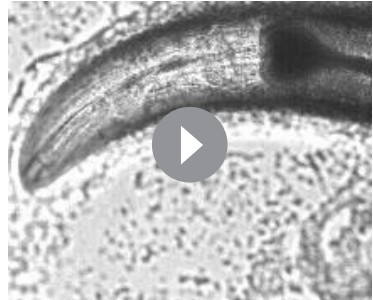

**Video 2.** Spitting pumps in polystyrene bead feeding assay. High frame rate video of *C. elegans* spitting in response to light in polystyrene bead feeding assay. During spitting pumps, beads no longer accumulate in the buccal cavity, but instead freely enter the pharyngeal lumen. At the end of the pump, all material ingested during the pump is expelled. Video was recorded at 1000 frames per second. Playback is at 3% of original speed.

https://elifesciences.org/articles/59341#video2

different motion: rather than relaxing and closing at each pump's conclusion, the anterior tip of the pharynx remained contracted—that is open—throughout the pumping cycle (*Figure 1F*; *Video 4*). This sustained contraction prevented the pharyngeal valve from closing and capturing food at the end of the pump, such that closure of the lumen expelled the pharyngeal contents. These open-valve pumping motions occurred specifically during the burst pumping phase, confirming that they are specific to spitting (*Figure 1—figure supplement 1B and C*).

In addition to increasing the rate of muscle contraction and preventing the closure of the pharyngeal valve, light altered a third aspect of pharyngeal pumping during the burst pumping phase: the metastomal filter, which is partially closed during feeding (*Fang-Yen et al., 2009*), instead appeared to be open during spitting. We inferred the configuration of the flaps of the metastomal filter based on the movement of beads through the buccal cavity – whereas during feeding pumps beads accumulated in the buccal cavity and filtered only gradually into the pharynx (*Figure 1C and G*; *Video 1*), during spitting pumps beads appeared to flow unrestricted from the buccal cavity into the pharynx (*Video 2*), indicating that the metastomal filter likely opens during spitting. We speculate that, in combination with the opening of the pharyngeal valve, this opening of the metastomal filter produces a rinsing effect, in which greatly increased quantities of material are drawn into and then expelled from the pharynx. Such a mechanism might act to rinse out harmful material before it is ingested (see Discussion).

Although incoming beads were typically impeded by the metastomal filter, we noticed that beads flushed freely into the pharynx during about 25% of feeding pumps; presumably during at least some of these pumps the metastomal filter was partially or fully opened (*Figure 1G*, *Video 5*). Because the amount of material ingested and retained during these pumps far exceeded the amount ingested and retained during most feeding pumps (*Figure 1—figure supplement 1D*), we designated such pumps as 'gulps' and propose that 'gulping' might be regulated to modulate rates of food ingestion.

In summary, we found that the spitting reflex was characterized by three changes to pharyngeal behavior: (a) the pumping rate increased; (b) the pharyngeal valve remained open instead of closing, producing spitting; and (c) the metastomal filter opened, increasing the number of particles rinsed in and out with each pump. These changes transformed the pharyngeal motor program from one that promotes feeding into one that promotes spitting from and rinsing of the procorpus (*Figure 1A and H*).

## Subcellularly localized muscle contractions and calcium increases open the pharyngeal valve to produce spitting

We next asked which muscles control the light-induced spitting reflex. The anterior pharynx contains three pharyngeal muscle classes: pm1, pm2, and pm3 (*Figure 2A*). pm1 is a single hexanucleate cell, while the pm2 and pm3 classes consist of three binucleate cells each (*Albertson and Thomson, 1976*). The pm1 and pm2 muscles, small cells of unknown function, form a ring around the anterior opening of the pharynx. The metastomal filter protrudes from this ring into the buccal cavity (*Figure 2A*) such that contraction of the radially arranged myofilaments of the pm1 and/or pm2 muscles would be expected to pull the filter open and thus expand the aperture of the anterior pharynx (*Figure 2A*). It is also conceivable that contraction of the pm2s could close the flaps in the absence of contraction by pm1 (*Figure 2A*). The pm3 muscles make up the remainder of the musculature of the procorpus, and one leaf of the pharyngeal valve protrudes from the anterior end of each of the pm3 muscle cells. Because the filter and valve are held open in spitting, we hypothesized that spitting involves the sustained contraction of some or all of pm1, the pm2s, and/or the anterior end of the pm3s.

We sought to determine the roles of these muscles in spitting. We laser-killed the pm1 and pm2 muscles singly and in combination to determine what, if any, function they have in spitting; because the pm3s are required for the structural integrity of the pharynx (Leon Avery, personal communication), pharyngeal function cannot be studied after pm3 ablation. Laser ablation of pm1, the pm2s, or both classes together did not eliminate burst pumping or spitting (*Figure 2B–E*; *Videos 6–9*), implying that the pm3s can hold open the pharyngeal valve and thus produce spitting in the absence of pm1 and the pm2s. Given that the pm3s are the only muscles in the vicinity of the valve once pm1 and the pm2s have been ablated, we conclude that the contraction of a subcellular region of the pm3 muscles must be sufficient to open the valve, that is the anterior regions of the pm3s of these

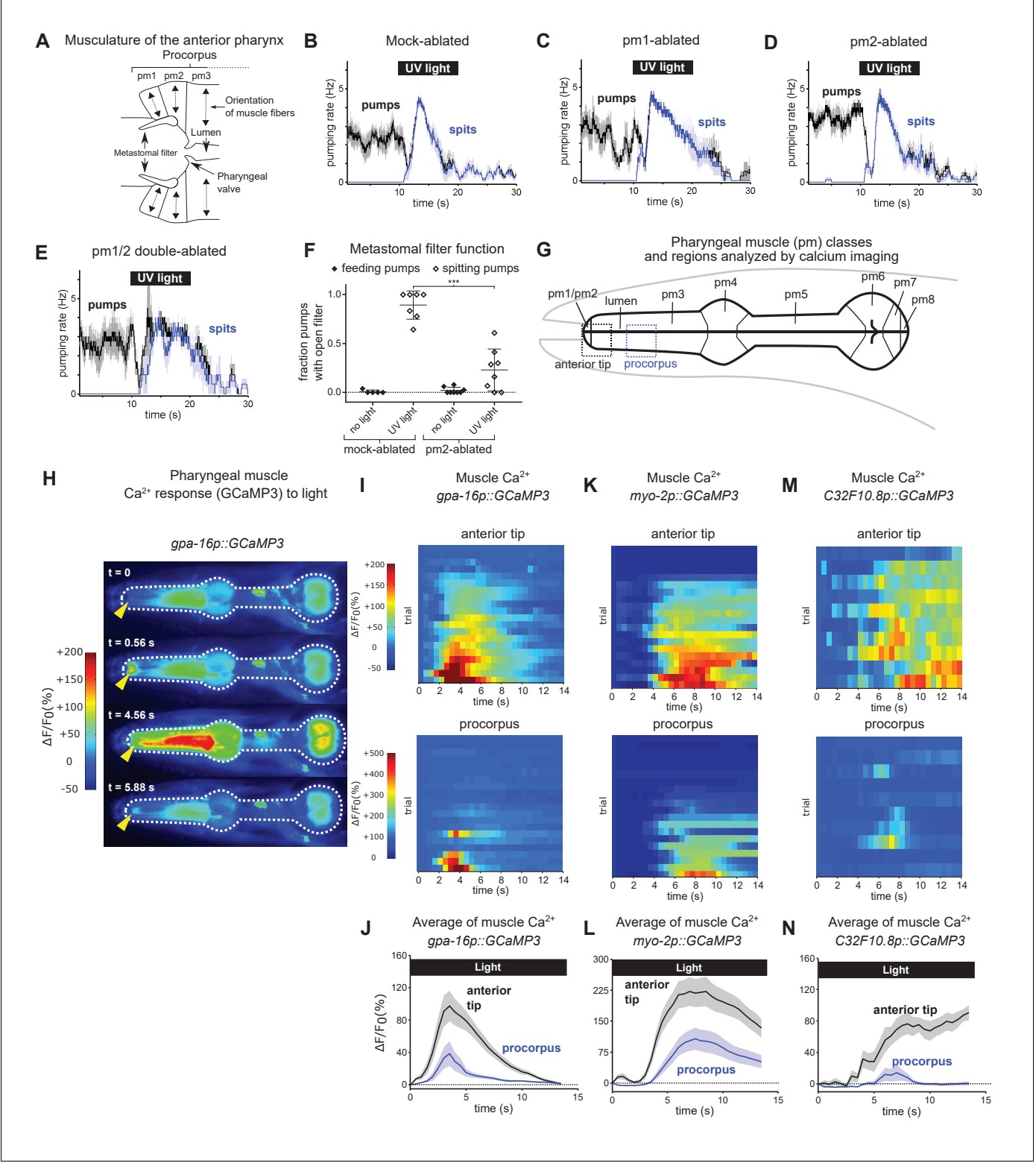

**Figure 2.** Subcellularly localized muscle contractions and calcium increases open the pharyngeal valve to produce spitting. (**A**) Anatomy of the anterior pharynx showing pharyngeal muscles pm1, pm2, and pm3. Double-headed arrows indicate orientation of muscle fibers. (**B–E**) Mock-ablated (n = nine animals), pm1-ablated (n = 5), pm2-ablated (n = 8), and pm1/pm2 double-ablated animals (n = 4) spit in response to light. All animals carried transgene *nls507 (pm1_promoter::gfp)* to mark pm1. '*pm1_promoter*' refers to the *inx-4* promoter. (**F**) Laser ablation of the pm2 pharyngeal muscles partially suppresses the light-induced opening of the metastomal filter. Center bar, mean; error bars, SD. ***, p < 0.001; t test. n ≥ five animals. (**G**) Diagram of the eight

*Figure 2 continued on next page*

*Figure 2 continued*

classes of pharyngeal muscle (pm1-8; borders indicated by thin solid black lines; lumen indicated with thick solid black lines). Dashed boxes indicate regions analyzed for GCaMP3 fluorescence changes. (H) Representative images from video of pharyngeal muscle GCaMP3 calcium imaging. Light induces subcellularly localized increases in pharyngeal muscle GCaMP fluorescence at the anterior end of the pharynx. Yellow arrowhead indicates spatially restricted calcium increases. Images false-colored based on the intensity of fluorescence changes. See also *Video 11*. (I) Pharyngeal muscle GCaMP responses of individual animals to light. Each row represents a different animal; $\Delta F/F_0$ over time is indicated according to the heatmap at left. Light induces subcellularly localized increases in pharyngeal muscle GCaMP fluorescence at the anterior end of the pharynx. Animals carried transgene *nIs686 (gpa-16p::GCaMP3)*, which expresses GCaMP in pharyngeal muscle under the *gpa-16* promoter. Each line indicates an independent trial from a different animal. Regions analyzed are indicated in (G). n = 20 animals. (J) Average GCaMP response of animals shown in (I). (K) GCaMP responses to light of pm3 muscles in individual animals. Light induces subcellularly localized increases in pm3 GCaMP fluorescence at the anterior end of pm3. Animals carried transgene *cuIs36 (myo-2p::GCaMP3)*, which expresses GCaMP in pm3 under the *myo-2* promoter. Each line represents an independent trial of a different animal. Regions analyzed are indicated in (G). n = 20 animals. (L) Average GCaMP response of animals shown in (K). (M) GCaMP responses to light of pm3 muscle cells in individual mosaic animals. Light induces subcellularly localized increases in pm3 GCaMP fluorescence at the anterior end of pm3. Animals carried transgene *nEx3045 (C32F10.8p::GCaMP3)*, which expresses GCaMP in pm3 under the *C32F10.8* promoter. Each line represents an independent trial of a different animal. Regions analyzed are indicated in (G). n = 10 animals. (N) Average GCaMP response of animals shown in (M). Shading around traces indicates SEM.

The online version of this article includes the following source data and figure supplement(s) for figure 2:

**Source data 1.** Source data for *Figure 2*.

**Figure supplement 1.** Additional characterization of the effects of pm1 ablation and of the immobilized muscle calcium imaging assay.

**Figure supplement 1—source data 1.** Source data for *Figure 2—figure supplement 1*.

**Figure supplement 2.** Characterization of the pharyngeal expression pattern of transgenes used in calcium imaging.

**Figure supplement 2—source data 1.** Source data for *Figure 2—figure supplement 2*.

animals underwent sustained contraction and held open the valve while the rest of the pm3s rhythmically contracted and relaxed.

While the pm1 and pm2 muscles were not necessary for spitting (i.e. the sustained opening of the pharyngeal valve), their ablation altered several aspects of pharyngeal function. First, pm2-ablated animals were partially defective in opening the metastomal filter—the filter often appeared to be closed during spitting (*Video 6*, *Figure 2F*). This abnormality might reflect a functional requirement for the pm2s in opening the flaps (as suggested by the anatomy) or an ablation-induced abnormality that reduced the mobility of the filter. Second, consistent with an earlier observation by others (L. Avery and C. Fang-Yen, personal communication), ablation of pm1 eliminated the metastomal filter (data not shown). For this reason, we were unable to determine whether pm1 is also involved in opening the filter. However, because pm2-ablated animals were not completely defective in opening the filter (*Figure 2F*) and because pm1 is the only other muscle positioned to perform this function, we think it likely that pm1 contributes to filter opening.

Consistent with a role for the metastomal filter in regulating particle influx, we observed that pm1- and pm1/2-ablated animals often ingested more beads into the procorpus than did mock-ablated animals (the 'stuffed pharynx' phenotype; *Avery and Horvitz, 1987*; *Figure 2—figure supplement 1A and B*). Some of these pm1- or pm1/2-ablated animals with full pharynges ejected beads less efficiently (five of nine animals; *Video 7*), requiring more pumps to spit out material than mock-ablated animals. The reduced spitting efficiency of these pm1-ablated animals might be a consequence of their enhanced ingestion of beads in the anterior pharynx—we also observed similarly impaired spitting in one of two mock-ablated animals that had spontaneously ingested greater than normal quantities of beads. However, because we cannot separate the effects of pm1 ablation on the metastomal flaps and particle intake from its effects on spitting behavior, we cannot exclude the possibility that pm1 functions in part to increase the efficiency of spitting. It is also possible that contractions of pm1 and/or pm2, despite being unnecessary for spitting given the results of our ablation studies, might be sufficient in non-ablated animals to open the pharyngeal valve even in the absence of pm3 contraction. With these *caveats*, our ablation experiments establish that, whatever role pm1 and the pm2s might play in other contexts, they are not required for light-induced spitting (i.e. opening of the pharyngeal valve).

We were surprised to observe a functional uncoupling of subdomains—the anterior end and the rest of the cell—within each of the pm3 muscle cells. We next sought the physiological mechanism that produces the sustained subcellular pm3 muscle contractions. Given the central role of

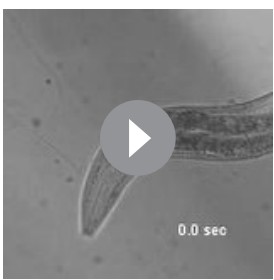

**Video 3.** Feeding pumps in muscle motion visualization assay. Video of *C. elegans* feeding beneath a coverslip. The pharyngeal lumen is visible as a white space when it is opened by the contraction of pharyngeal muscle. In feeding, the musculature of the anterior pharynx relaxes at the end of each pump, closing the pharyngeal valve and thereby trapping ingested material. Animals are slow-moving *unc-29 (e1072)* mutants to facilitate video capture. Video was recorded at 86 frames per second. Playback is at 50% of original speed.
https://elifesciences.org/articles/59341#video3

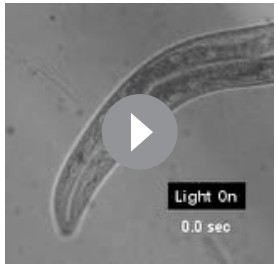

**Video 4.** Spitting pumps in muscle motion visualization assay. Video of *C. elegans* beneath a coverslip spitting in response to light. In contrast to feeding pumps, spitting pumps are characterized by the sustained contraction of the anterior musculature of the pharynx, which remains contracted open as the rest of the pharynx continues to contract and relax. Animals are slow-moving *unc-29(e1072)* mutants to facilitate video capture. Video was recorded at 86 frames per second. Playback is at 50% of original speed.
https://elifesciences.org/articles/59341#video4

intracellular calcium in initiating muscle contraction (*Kuo and Ehrlich, 2015*), we examined intracellular calcium levels by expressing the calcium reporter GCaMP3 (*Tian et al., 2009*) in pharyngeal muscle and imaging light-induced fluorescence calcium changes in the pharynges of immobilized animals (*Kim et al., 2013a*). These imaging conditions differed from those we used to analyze spitting in the free-moving assay; for imaging, animals were immobilized and this immobilization suppressed spontaneous pumping. Nonetheless, light often stimulated burst pumping by immobilized animals (*Figure 2—figure supplement 1C and D* and see Materials and methods below), and these light-stimulated burst pumps were dependent on the M1 neuron (*Figure 2—figure supplement 1D*), which is necessary for spitting (*Bhatla et al., 2015*), and resulted in the expulsion of previously ingested mineral oil, as is the case for spitting by free-moving animals (*Video 10*). Taken together, these results indicate that the light-induced pumps of immobilized animals are spits.

We first used the *gpa-16* promoter (*Jansen et al., 1999*) to broadly drive the expression of GCaMP3 in pharyngeal muscle. We characterized the expression pattern of this transgene using confocal microscopy and observed bright fluorescence in the pm2 and pm3 muscles and also in the mc1 marginal cells (*Figure 2—figure supplement 2A*). We also observed occasional dim fluorescence in pm1 (3 of 10 animals). After stimulating animals with light, we often detected pumps as calcium increases across the procorpus and the anterior and terminal bulbs, indicating that functional GCaMP was present in these regions (*Video 11*, *Figure 2G–H*). Interestingly, calcium signals at the tip of the pharynx began before and persisted after calcium signals in posterior regions (*Figure 2H–J*). Additionally, during many trials (12 of 20) this anterior region responded with a sustained rise in calcium even when the rest of the pharynx did not respond (*Figure 2I*). These calcium responses to light were imaged at an acquisition rate of 2 Hz, so we lacked the temporal resolution to distinguish whether each localized calcium signal was a single sustained increase or a rapid series of short-duration calcium pulses. The location and temporal dynamics of the sustained calcium signals in the anterior pharynx coincided precisely with those of the

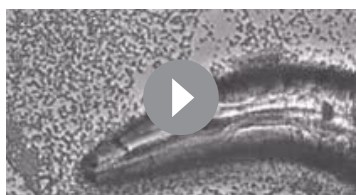

**Video 5.** 'Gulping' pumps in polystyrene bead feeding assay. High frame rate video of *C. elegans* 'gulping' in polystyrene bead feeding assay. As in spitting pumps, beads pass freely through the buccal cavity and into the pharyngeal lumen. As in feeding pumps, this material is subsequently trapped and retained by the pharyngeal valve, resulting in ingestion. Video was recorded at 1000 frames per second. Playback is at 3% of original speed.
https://elifesciences.org/articles/59341#video5

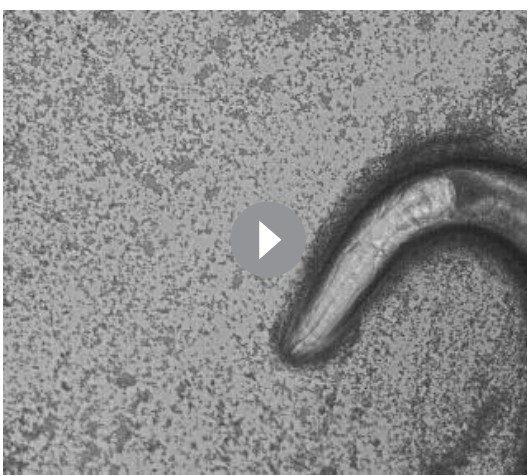

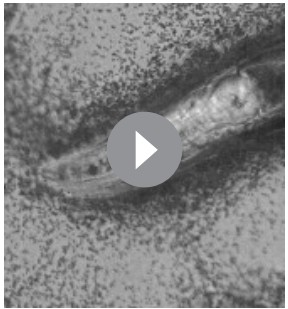

**Video 6.** pm2-ablated animals are defective in opening the metastomal filter during spitting behavior in polystyrene bead feeding assay. High frame rate video of pm2-ablated *C. elegans* spitting in response to light in polystyrene bead feeding assay. In contrast to typical spitting pumps, beads accumulate in the buccal cavity, indicating that the metastomal filter is partially closed. At the end of the pump, ingested material is still expelled, indicating that the pharyngeal valve is open. Animals carried transgene *nIs507 [inx-4*<sub>promoter</sub>*::gfp; lin-15(+)]*, which was used to mark the pm1 muscle for ablation. Video was recorded at 1000 frames per second. Playback is at 3% of original speed.
https://elifesciences.org/articles/59341#video6

**Video 7.** A subset of pm1-ablated animals spit less efficiently. High frame rate video of pm1-ablated *C. elegans* spitting in response to light in polystyrene bead feeding assay. At the end of the pump, ingested material is expelled, indicating that the pharyngeal valve is open, but material already held in procorpus is ejected less efficiently in comparison to mock-ablated animals. Animals carried transgene *nIs507 [inx-4*<sub>promoter</sub>*::gfp; lin-15(+)]*, which was used to mark the pm1 muscle for ablation. Video was recorded at 1000 frames per second. Playback is at 3% of original speed.
https://elifesciences.org/articles/59341#video7

sustained contraction of the muscle region surrounding the anterior pharyngeal valve that underwent sustained contraction (i.e., became uncoupled from the 4–5 Hz rate of pumping) during spitting behavior. Consistent with previous reports (*Kerr et al., 2000*; *Shimozono et al., 2004*), we did not observe spatially restricted calcium signals in the anterior pharynx during feeding pumps stimulated with serotonin, indicating that these localized calcium transients are specific to spitting (*Video 12*, *Figure 2—figure supplement 1E*).These observations strongly suggest that these localized calcium increases act physiologically to drive the subcellular contraction of the anterior ends of the pm3s.

To determine if the calcium transients we observed were restricted cell-specifically to the pm1 and/or pm2 muscles or whether they reflected subcellularly localized activity in the pm3s and/or mc1s, we used two different promoters that each drive expression specifically in the pm3s and/or mc1s but not in pm1 or the pm2s.

The *myo-2p* promoter drives expression in the pm3s and in no other cell in the anterior pharynx (*Okkema et al., 1993*; *Okkema and Fire, 1994*). We obtained animals that express GCaMP3 under this promoter (*Kozlova et al., 2019*) and used confocal microscopy to confirm that these *myo-2p:: GCaMP3* animals expressed GCaMP in the pm3s but not in pm1 or the pm2s (*Figure 2—figure supplement 2B*). After stimulation with light, 16 of the 19 *myo-2p::GCaMP* animals examined responded with subcellular calcium responses at the anterior tips of the pm3s (*Video 13*); 11 of these 16 animals also showed a subsequent calcium increase in the posterior *myo-2*-expressing regions of the pharynx (*Figure 2K–L*). We did not observe localized fluorescence signals when we imaged animals expressing a *myo-2p::GFP* transgene (*Miedel et al., 2012*) that drives expression of GFP in pharyngeal muscle (*Figure 2—figure supplement 1F*), indicating that the localized GCaMP3 fluorescence increases we observed were not a motion artefact or other calcium-independent effect. These observations demonstrate that light induces subcellularly localized calcium increases in the anterior ends of the pm3s. Unlike other strains tested in this study, immobilized animals carrying the *myo-2p::GCaMP3* transgene did not pump when stimulated with light (*Figure 2—figure supplement 1D*; see Material and methods below). The *myo-2* promoter used in this transgene drives transgene expression more strongly than the other pharyngeal muscle promoters used in this study

(i.e. *gpa-16p* and *C32F10.8p*), and the failure of animals expressing *myo-2p::GCaMP3* to pump while immobilized might be a consequence of GCaMP-mediated intracellular calcium-buffering (a known artefact of GCaMP overexpression—e.g. as noted by *Singh et al., 2018*) or another abnormality induced by transgene overexpression.

To confirm the conclusion that light induces subcellularly localized increases in the anterior ends of the pm3s, we used a second promoter to express GCaMP in the pm3s but not in pm1 or the pm2s, *C32F10.8p*. We identified *C32F10.8p* using confocal microscopy to screen GFP fluorescent transcriptional reporters of genes reported to be expressed in the muscles of the anterior pharynx; *C32F10.8p::GFP* (*McKay et al., 2003*; *Hunt-Newbury et al., 2007*) was expressed in both the pm3s and mc1s but not in pm1 or the pm2s (*Figure 2—figure supplement 2C*). We used the *C32F10.8* promoter to drive expression of GCaMP3 and confirmed using confocal microscopy that this expression was specific to the pm3s and mc1s (*Figure 2—figure supplement 2D*). Because of the occasional loss of extrachromosomal transgene arrays during cell division, *C. elegans* extrachromosomal arrays are expressed mosaically and the specific subset of cells that inherit the array varies from animal to animal (*Stinchcomb et al., 1985*). We selected and imaged mosaic *C32F10.8p::GCaMP3* transgene-positive animals with bright transgene expression in the anterior pharynx and observed localized fluorescence increases in response to light in the anterior pharynges of these animals (*Figure 2—figure supplement 1G*), showing that a GCaMP response to light occurs subcellularly in the anterior ends of the pm3s and/or mc1s. To clarify whether the pm3s, mc1s or both respond to light, we identified transgene-bearing mosaic animals that sparsely but brightly expressed GCaMP3 in one or several cells in the procorpus, examined those animals using our calcium-imaging assay, and then determined their transgene expression pattern using confocal microscopy. By combining pm3- and mc1-identifying landmarks in our calcium imaging videos with 3D reconstructions of GCaMP3 expression, we were able to obtain videos from which we could confidently determine the identities of specific cells that responded to light. Of the 39 videos we recorded, 12 showed cells we unambiguously identified as pm3, and 10 of these 12 videos (~83%) showed subcellular responses in the anterior end of that pm3 (*Figure 2M–N*; *Video 14*). Of the seven videos in which we could unambiguously identify the anterior end of an mc1 marginal cell, only one showed a calcium increase. The response of the set of 10 cells unambiguously identified as pm3s (*Figure 2M–N*) was similar to the response of the set of 10 unidentified cells we imaged in animals selected based on general procorpus expression (i.e. the set described in *Figure 2—figure supplement 1G*). These observations demonstrate that the pm3s

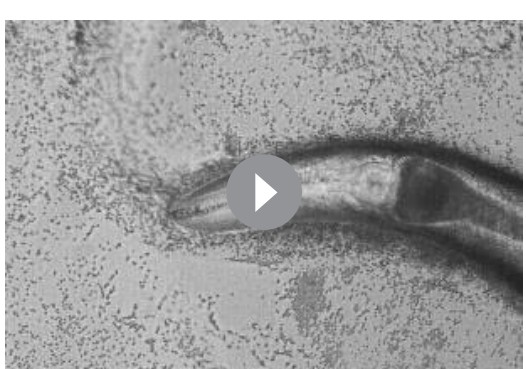

**Video 8.** A subset of pm1-ablated animals exhibit normal spitting behavior. High frame rate video of pm1-ablated *C. elegans* spitting in response to light in polystyrene bead feeding assay. Spitting behavior is comparable to that of mock-ablated animals. Animals carried transgene *nIs507 [inx-4_{promoter}::gfp; lin-15(+)]*, which was used to mark the pm1 muscle for ablation. Video was recorded at 1000 frames per second. Playback is at 3% of original speed.
https://elifesciences.org/articles/59341#video8

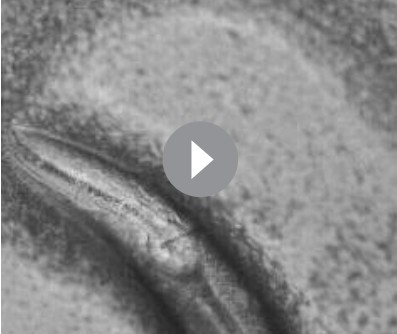

**Video 9.** pm1/pm2 double-ablated animals can hold open the pharyngeal valve and spit. High frame rate video of pm1/pm2 double-ablated *C. elegans* spitting in response to light in polystyrene bead feeding assay. At the end of the pump, the material ingested during the pump is spat out, indicating that the pharyngeal valve is open and that animals lacking both pm1 and the pm2s can spit. Animals carried transgene *nIs507 [inx-4_{promoter}::gfp; lin-15(+)]*, which was used to mark the pm1 muscle for ablation. Video was recorded at 1000 frames per second. Playback is at 3% of original speed.
https://elifesciences.org/articles/59341#video9

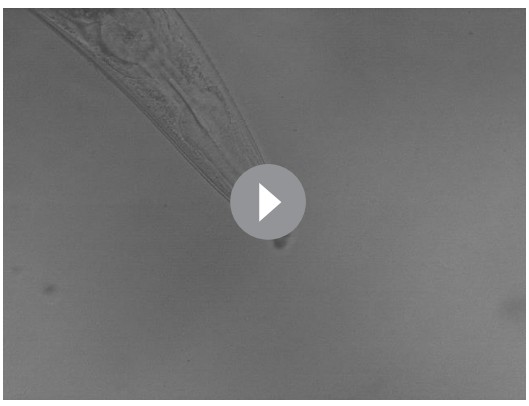

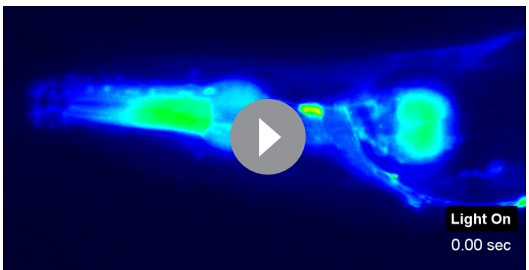

**Video 10.** Light induces immobilized animals to spit. Video of immobilized *C. elegans* spitting in response to 2 Hz flickered 365/485 nm light (i.e. the same as delivered to animals during calcium imaging). Animal was selected based on its ingestion of mineral oil into the anterior pharynx. Oil is visible as dark liquid that is ejected from the anterior pharynx over the course of several pumps. Animals carried the calcium-imaging transgene *nIs686 [gpa-16p::gcamp3; lin-15(+)]*. Video was recorded at 50 frames per second. Playback is at 100% of original speed.

https://elifesciences.org/articles/59341#video10

**Video 11.** Pharyngeal muscle calcium response to light (*gpa-16p::gcamp3* transgene). The pharyngeal muscle calcium response to light, as reported by fluorescence of GCaMP3 transgene *nIs686* (*gpa-16p::gcamp3*), which is expressed in the pm2 and pm3 pharyngeal muscles, the mc1 marginal cells, and occasionally/dimly in the pm1 pharyngeal muscle. Pumps result in fluorescence increases across the entirety of the anterior pharynx. Before and between pumps, localized and sustained calcium increases occur at the anterior tip of the pharynx. Animals carried the calcium-imaging transgene *nIs686 [gpa-16p::gcamp3; lin-15(+)]*. Video was recorded at 15 frames per second and false-colored based on the intensity of fluorescence changes (blue, low intensity; red, high intensity). Playback is at 100% of original speed.

https://elifesciences.org/articles/59341#video11

responded to light with robust subcellularly localized calcium increases. The mc1s might also exhibit a response that is less frequent, less intense, and/or at least partially outside of the dynamic range of GCaMP3.

In conclusion, our imaging of strains carrying three different reporters—*gpa-16p::GCaMP3*, *myo-2p::GCaMP3*, and *C32F10.8p::GCaMP3*—establish that a subcellular compartment of the anterior tip of the pm3s undergoes a sustained calcium increase in response to light. Taken together, our results indicate that the pm1, pm2, and pm3 muscles play specialized roles in controlling pharyngeal particle flow during spitting: the pm2 and likely the pm1 muscles actively open the metastomal filter, increasing the aperture size of the anterior lumen and thus facilitating particle influx, while a subcellular anterior compartment of the pm3s opens the pharyngeal valve via subcellular calcium signaling, preventing the retention of ingested material and thus producing spitting.

### Pharyngeally expressed gustatory receptor orthologs *lite-1* and *gur-3* play major and minor roles, respectively, in the light-induced spitting reflex

Previously-described *C. elegans* locomotory and pharyngeal feeding responses to light depend on the gustatory receptor orthologs *lite-1* and/or *gur-3*, which are expressed in a number of pharyngeal and extra-pharyngeal neurons (*Edwards et al., 2008*; *Liu et al., 2010*; *Bhatla and Horvitz, 2015*). We asked if gustatory receptor orthologs similarly control the light-induced spitting reflex. The burst pumping rates of animals carrying null alleles of *gur-3* (*Bhatla and Horvitz, 2015*) or its paralogs *egl-47/gur-1*, *gur-4,* and *gur-5* were similar to that of wild-type animals (*Figure 3—figure supplement 1A–E*). *lite-1* null mutants (*Edwards et al., 2008*) could not be tested directly, because they exhibit substantial defects in light-induced pumping inhibition (*Bhatla and Horvitz, 2015*), thus obscuring burst pumping behavior (*Figure 3—figure supplement 1F and G*).

To examine the role of *lite-1* in burst pumping, we tested *lite-1* mutants in an *eat-2* mutant background. *eat-2* mutants are severely defective in feeding pumping (*Raizen et al., 1995*) but not in burst pumping (*Figure 3—figure supplement 1H*). Thus, the burst pumping rate rises above the slow *eat-2* feeding pumping rate and can be scored even in animals defective in light-induced

inhibition of feeding. We found that in *eat-2* mutants *lite-1* but not *gur-3* was required for wild-type burst pumping, a defect that was partially rescued by expression of a wild-type copy of *lite-1* (*Figure 3—figure supplement 1H–L*). Thus, light-induced stimulation of pumping requires *lite-1* but not *gur-3*.

These experiments established a role for *lite-1* in controlling burst pumping—that is, light-induced contractions of the grinder. To determine directly whether *lite-1* and/or *gur-3* are required for spitting—that is, the opening of the pharyngeal valve—we used the polystyrene bead-feeding assay to measure light-induced spitting (i.e. the failure to capture beads at the conclusion of a pump) by *lite-1* and *gur-3* single mutants and *lite-1 gur-3* double mutants. Both *gur-3* and *lite-1* mutants spit less than controls, and *lite-1* mutants also displayed a dramatic reduction in peak spitting rate compared to wild-type animals (*Figure 3A–C and G*). Strikingly, *lite-1 gur-3* double mutants did not spit in response to light (*Figure 3D*). *lite-1* and *gur-3* mutants were also defective in opening the metastomal filter (*Figure 3H*). Rescue transgenes expressing wild-type copies of either *lite-1* (*Bhatla and Horvitz, 2015*) or *gur-3* (this study) restored both light-induced spitting and filter-opening in *lite-1 gur-3* double mutants (*Figure 3F–H*). Together, these results indicate that both *lite-1* and *gur-3* function in light-induced spitting and opening of the filter, with *lite-1* likely playing a greater role.

Next, we assayed the pharyngeal muscle calcium response to light in *lite-1* and *gur-3* mutants using the pan-pharyngeal *gpa-16p::GCaMP3* reporter. *gur-3* mutants resembled wild-type animals (*Figures 3I, J and M*), while *lite-1* mutants were almost completely defective in the pharyngeal muscle calcium response to light, except for a very weak, fast response (*Figures 3I, K, M and N*). This fast response depended on *gur-3*, as the response of *lite-1 gur-3* double mutants was completely defective (*Figures 3I, L and N*). Thus, consistent with our behavioral observations, *lite-1* and *gur-3* function together in light-induced calcium increases in pharyngeal muscle, with *lite-1* playing the major role and *gur-3* playing a minor role.

Taken together, these results indicate that *lite-1* and *gur-3* both contribute to light-induced spitting. *lite-1* is necessary for the full response—an increase in pumping rate (burst pumping), opening of the pharyngeal valve (i.e. spitting), opening of the metastomal filter, and a robust increase in pharyngeal muscle calcium—while *gur-3* is sufficient in the absence of *lite-1* to produce an attenuated response—partial opening of the valve and filter and weak activation of pharyngeal muscle.

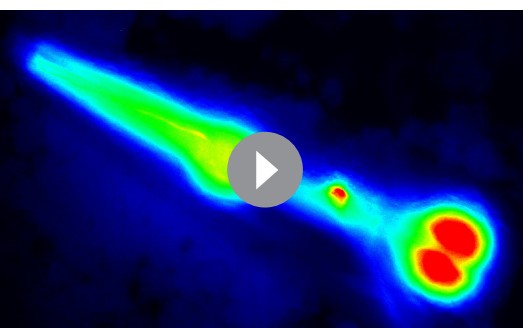

**Video 12.** Sustained, anteriorly localized calcium signals do not occur when animals are induced to pump using serotonin. Animals carrying the muscle-GCaMP transgene *nIs686 [gpa-16p::gcamp3; lin-15(+)]* were immobilized in the presence of 10 mM serotonin and filmed at the onset of pumping. Video was recorded at four frames per second and false-colored based on the intensity of fluorescence changes (blue, low intensity; red, high intensity). Playback is at 100% of original speed.
https://elifesciences.org/articles/59341#video12

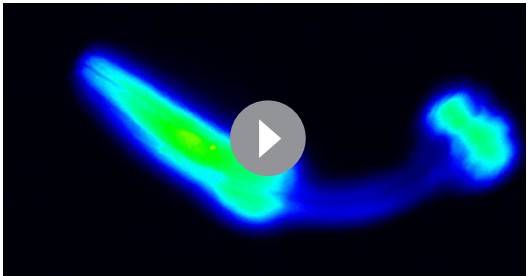

**Video 13.** Pharyngeal muscle calcium response to light (*myo-2p::gcamp3* transgene). The pharyngeal muscle calcium response to light as reported by fluorescence of GCaMP3 transgene *cuIs36* (*myo-2p::gcamp3*), which is expressed specifically in the pm3s but not the pm2s and pm1. Localized and sustained calcium increases occur at the anterior tip of the pharynx. Video was recorded at two frames per second and false-colored based on the intensity of fluorescence changes (blue, low intensity; red, high intensity). Playback is at 100% of original speed.
https://elifesciences.org/articles/59341#video13

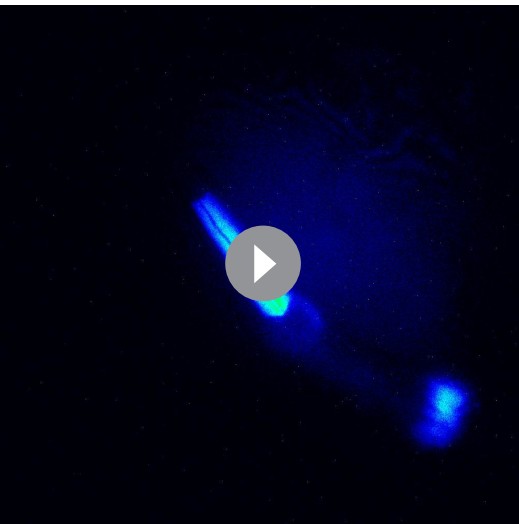

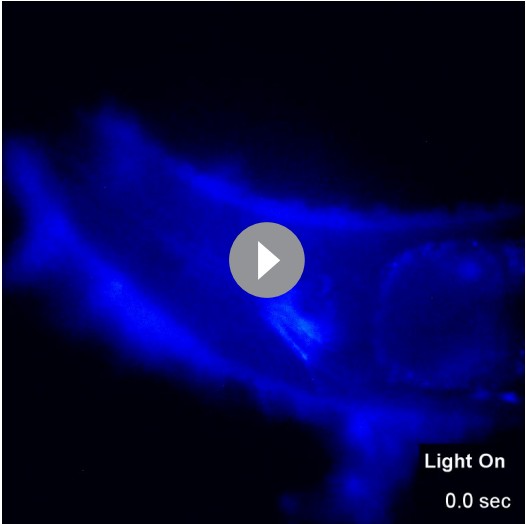

**Video 14.** Pharyngeal muscle calcium response to light (*C32F10.8p::gcamp3* transgene). The pharyngeal muscle calcium response to light, as reported by fluorescence of GCaMP3 transgene *nEx3045 [C32F10.8p::gcamp3; lin-15(+)]*, which is expressed specifically in the pm3 pharyngeal muscles and mc1 marginal cells. Localized and sustained calcium increases occur at the anterior tip of the pharynx. Video was recorded at two frames per second and false-colored based on the intensity of fluorescence changes (blue, low intensity; red, high intensity). Playback is at 100% of original speed.

https://elifesciences.org/articles/59341#video14

**Video 15.** The pharyngeal neuron M1 is activated by light (*glr-2p::gcamp6s* transgene). The pharyngeal neuron M1 is acutely activated by light, as reported by fluorescence of GCaMP6s calcium imaging transgene *nIs678 [glr-2p::gcamp6s; lin-15(+)]*. Video was recorded at two frames per second and false-colored based on the intensity of fluorescence changes (blue, low intensity; red, high intensity). Playback is at 100% of original speed.

https://elifesciences.org/articles/59341#video15

## The M1 pharyngeal neuron innervates the pharyngeal valve and controls light-induced spitting

The M1 pharyngeal neuron is the only major source of innervation of the pm1, pm2, and pm3 muscles at the anterior tip of the pharynx (*Albertson and Thomson, 1976*), and these M1 synapses are located in the region of the metastomal filter, the pharyngeal valve at the anterior corpus, the subcellular muscle contractions and the subcellular calcium increases we observed during spitting (*Figure 4A*). We previously found that M1 is required for spitting induced by violet light (*Bhatla et al., 2015*). We tested whether M1 is also required for spitting induced by UV light, which, as noted above, we used throughout this study to induce spitting. We laser-killed M1 and also generated an M1 genetic-ablation strain in which the mammalian caspase ICE (*Cerretti et al., 1992*; *Thornberry et al., 1992*; *Zheng et al., 1999*) is expressed under the M1-specific *lury-1* promoter (*Ohno et al., 2017*). Unlike mock-ablated or transgene-negative control animals, animals lacking M1 because of laser or genetic ablation were completely defective in UV-light-induced burst pumping and spitting (*Figure 4B–E* and *Figure 4—figure supplement 1A–C*). Genetic ablation of M1 also eliminated burst pumping in *eat-2* mutants (*Figure 4—figure supplement 1D*). Finally, laser ablation of M1 in animals carrying the broadly-expressed *gpa-16p::GCaMP3* reporter also eliminated the light-induced calcium signals in anterior pharyngeal muscles (*Figure 4F–G*), consistent with their being induced by M1. A *caveat* concerning this interpretation is that ablation of M1 might also have affected the development or function of other pharyngeal components, but we note that when we previously laser-killed each pharyngeal neuron class individually and tested the response of these ablated animals to light, only ablation of M1 caused a reduction in burst pumping (*Bhatla et al., 2015*). We could not directly determine whether ablation of M1 altered the function of the metastomal filter in wild-type animals, because M1-ablated animals did not spit and rarely pumped in the presence of light (but see below). To determine if M1 activity is acutely required for spitting, we generated animals expressing the HisCl1 histamine-gated chloride channel (*Pokala et al., 2014*) in M1 using an extrachromosomal array that drives *HisCl1* gene expression by the *glr-2* promoter

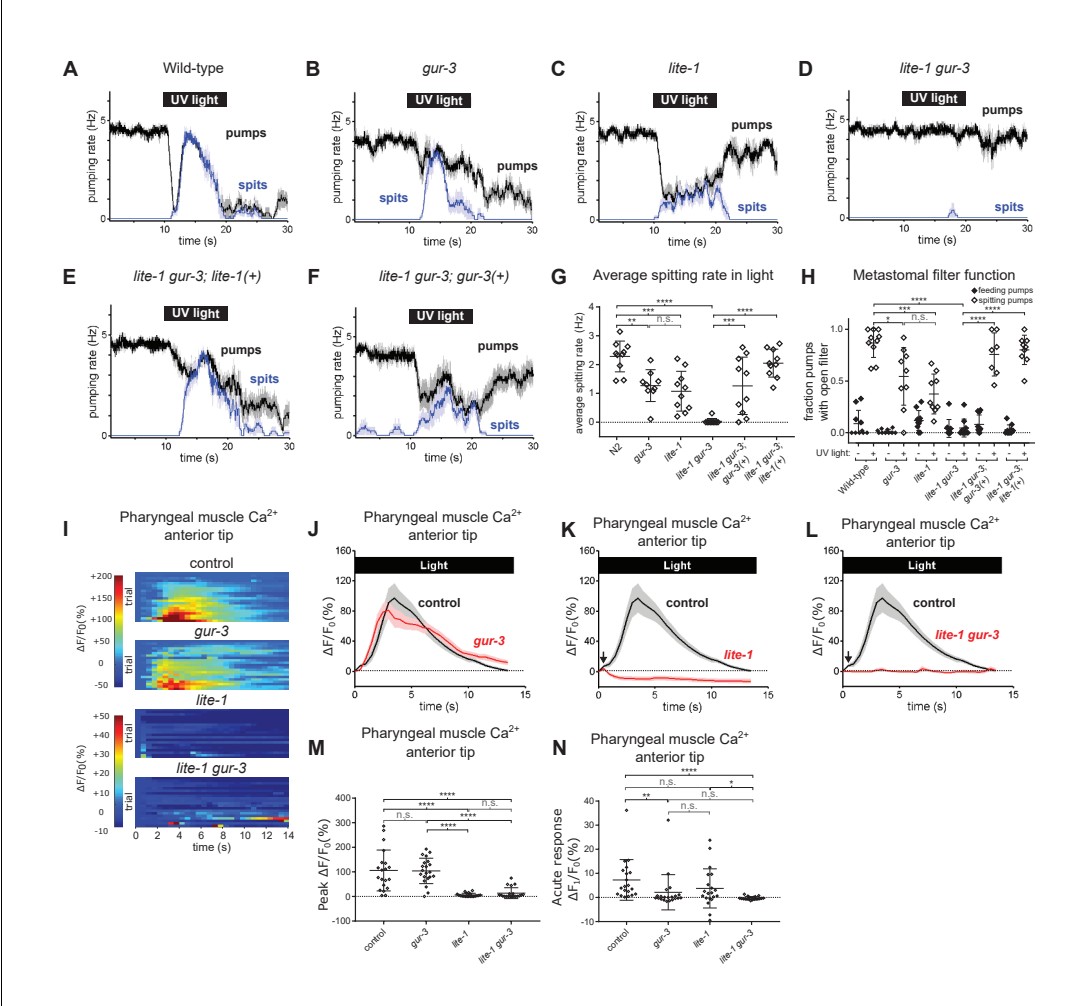

**Figure 3.** Pharyngeally expressed gustatory receptor orthologs *lite-1* and *gur-3* play major and minor roles, respectively, in the light-induced spitting reflex. (A–D) Light-induced spitting is reduced in *gur-3* and *lite-1* single mutants and eliminated in *lite-1 gur-3* double mutants. n = 10 animals except n$_{gur-3}$ = 9. (E–F) Expression of wild-type *lite-1* (transgene *nEx2281; lite-1$_{promoter}$::lite-1::gfp*) or *gur-3* (transgene *nEx2157; gur-3$_{promoter}$::gur-3 gDNA*) respectively restores wild-type or partial spitting to *lite-1 gur-3* double mutants. n = 10 animals. (G) Quantification of average spitting rate during light exposure of animals from (H–M). ****, approximate adjusted p value < 0.0001; ***, adjusted p value < 0.0005, **, adjusted p value < 0.005; n.s., not significant; ordinary one-way ANOVA. (H) Quantification of the frequencies at which the metastomal filter was open in feeding and spitting pumps in (H–M). ****, approximate adjusted p value < 0.0001; ***, adjusted p value < 0.0005, *, adjusted p value < 0.05; n.s., not significant; ordinary one-way ANOVA. (I) Pharyngeal muscle GCaMP responses of individual animals to light. Each row represents a different animal; ΔF/F$_0$ over time is indicated according to the heatmap at left. n ≥ 20 animals. Control data duplicated from *Figure 2I–J*. (J–L) Light-induced pharyngeal muscle calcium increases depend on *lite-1* (which accounts for 99% of the response) and *gur-3* (which makes a small contribution to the initial activation of pharyngeal muscle; black arrows). n ≥ 20 animals. Control data duplicated from *Figure 2I–J*. (M) Quantification of peak GCaMP ΔF/F$_0$ from (Q–T). *lite-1*, but not *gur-3*, is required for the amplitude of the pharyngeal muscle response to light. ****, approximate adjusted p value < 0.0001; n.s., not significant; Kruskal-Wallis test, followed by Dunn's multiple comparison test. (N) Quantification of acute (F$_{t = 0.5 sec}$/F$_0$) GCaMP response from (Q–T). *gur-3*, but not *lite-1*, is required for the acute GCaMP response of pharyngeal muscle to light. ****, approximate adjusted p value < 0.0001; **, adjusted p value < 0.001; *, adjusted p value < 0.05; n.s., not significant; Kruskal-Wallis test, followed by Dunn's multiple comparison test. Shading around traces indicates SEM. All *gur-3* alleles are null allele *gur-3(ok2245)* and all *lite-1* alleles are null allele *lite-1(ce314)*. Center bar, mean; error bars, SD. ****, approximate adjusted p value < 0.0001; ***, adjusted p value < 0.0005, **, adjusted p value < 0.005; *, adjusted p value < 0.05; n.s., not significant; ordinary one-way ANOVA. The online version of this article includes the following source data and figure supplement(s) for figure 3:

**Source data 1.** Source data for *Figure 3*.

**Figure supplement 1.** Additional characterization of gustatory receptor ortholog function and expression.

**Figure supplement 1—source data 1.** Source data for *Figure 3—figure supplement 1*.

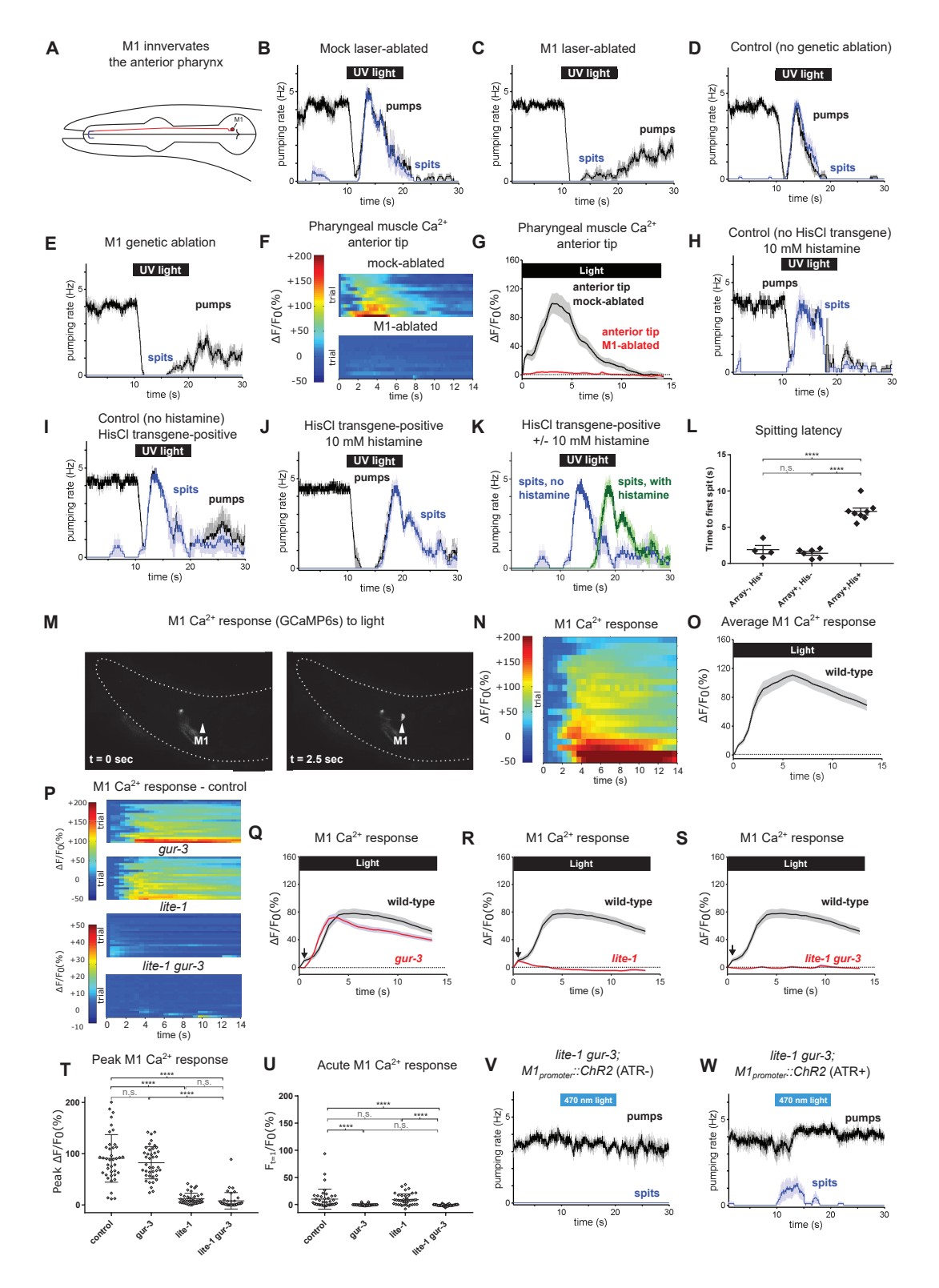

**Figure 4.** The pharyngeal neuron M1 controls the spitting reflex. (**A**) The anatomy of M1 (red) suggests that it controls spitting directly, as it is the only neuron to make neuromuscular junctions (bolded blue segment) with the region of pharyngeal muscle that undergoes sustained contraction during spitting. (**B–C**) M1 laser-ablated, but not mock-ablated, animals are completely defective in light-induced spitting and burst pumping. Animals carried GFP reporter transgene *nls864 (M1_promoter::gfp)* to mark M1. '*M1_promoter*' refers to the *glr-2* promoter. n ≥ 12 animals. (**D–E**) M1 genetically-ablated

*Figure 4 continued on next page*

Figure 4 continued

animals, but not transgene-negative sibling controls, are completely defective in light-induced spitting and burst pumping. Ablated animals carried genetic-ablation transgene nEx2905 (M1$_{promoter}$::ice::sl2::mcherry) and all animals carried GFP reporter transgene nIs864. 'M1$_{promoter}$' refers to the lury-1 promoter. n = 10 animals. (F) Pharyngeal muscle GCaMP responses of individual animals to light. Each row represents a different animal; ΔF/F$_0$ over time is indicated according to the heatmap at left. n ≥ 11 animals. (G) Average response of animals from (F). Spatially restricted calcium signals depend on M1. 'Anterior tip' region analyzed is indicated in *Figure 2G*. Animals carried pharyngeal muscle GCaMP3 transgene nIs686 (gpa-16p::GCaMP3). (H–J) Histamine-exposed animals expressing the inhibitory histamine-gated chloride channel HisCl1 in M1 (n = 8) but neither of histamine-exposed animals lacking HisCl1 expression in M1 (n = 4) or non-histamine-exposed animals expressing HisCl1 in M1 (n = 6) exhibited significantly delayed spitting in response to light. (K) Comparison of the light-induced spitting behavior of animals from (I–J). (L) Quantification of H-J. Histamine-exposed animals expressing HisCl1 in M1 exhibit significantly greater spitting latencies than control animals. ****, adjusted p value < 0.0001; n.s., not significant; ordinary one-way ANOVA. (M) Representative example of light-induced GCaMP6s fluorescence increase in M1. Animal carried transgene nIs678 (M1$_{promoter}$:: gcamp6s). 'M1$_{promoter}$' refers to the glr-2 promoter. The unchanging signal common to both images was caused by GCaMP6s fluorescence in additional glr-2-expressing neurons in the nerve ring. See also *Video 15*. (N) M1 somatic GCaMP responses of individual animals to light; n = 20. (O) Average response of animals shown in (N). (P) M1 somatic GCaMP responses of individual control, gur-3, lite-1, and lite-1 gur-3 mutant animals to light. n = two biological replicates (40 animals total). (Q–S) Light-induced calcium increases in the M1 neuron depend on lite-1 (which accounts for 99% M1's response) and gur-3 (which makes a small but consistent contribution to M1's initial activation; black arrows). n = two biological replicates (40 animals total). (T) Quantification of peak GCaMP ΔF/F$_0$ from (P–S). lite-1, but not gur-3, is required for the amplitude of the M1 GCaMP response to light. ****, approximate adjusted p value < 0.0001; n.s., not significant; Kruskal-Wallis test, followed by Dunn's multiple comparison test. (U) Quantification of acute (F$_{t = 0.5 sec}$/F$_0$) GCaMP response from (P–S). gur-3, but not lite-1, is required for the acute M1 GCaMP response to light. ****, approximate adjusted p value < 0.0001; n.s., not significant; Kruskal-Wallis test, followed by Dunn's multiple comparison test. (V–W) A subset (~25%) of lite-1 gur-3 animals expressing an M1-specific ChR2 transgene nEx2815 (M1$_{promoter}$::chr2::sl2::mcherry) grown in the presence, but not the absence, of ATR spit in response to 470 nm light. 'M1$_{promoter}$' refers to the lury-1 promoter. n ≥ 10 animals. Shading around traces indicates SEM. All gur-3 alleles are null allele gur-3 (ok2245), and all lite-1 alleles are null allele lite-1(ce314).

The online version of this article includes the following source data and figure supplement(s) for figure 4:

**Source data 1.** Source data for *Figure 4*.
**Figure supplement 1.** Additional characterization of the function of the M1 pharyngeal neuron in spitting and burst pumping.
**Figure supplement 1—source data 1.** Source data for *Figure 4—figure supplement 1*.
**Figure supplement 2.** Burst pumping responses of acetylcholine receptor mutants to UV light.
**Figure supplement 2—source data 1.** Source data for *Figure 4—figure supplement 2*.

(*Brockie et al., 2001*), which overlaps with the M1-specific lury-1 promoter only in M1, and hyperpolarized M1 with exogenous histamine. Consistent with an acute function for M1 in spitting, exogenous histamine significantly delayed light-induced spitting in animals with array-expressing M1s, but not animals with array-negative M1s (*Figure 4H–L*). We conclude that the M1 neuron is required for UV light-induced burst pumping, spitting, and muscle calcium physiology.

We observed that the pumping rate of M1-ablated animals recovered significantly more quickly than did controls after stimulation with light and that these pumps, in contrast to typical pumps in which the procorpus and grinder contract together, were characterized by contractions of the grinder but not the procorpus (*Figure 4—figure supplement 1E–H*). These observations suggest that in addition to controlling pumping and spitting in the burst pumping phase, M1 also inhibits the grinder during the recovery phase.

Given that light-induced spitting requires M1, we asked whether M1 is acutely activated by light. We generated transgenic animals expressing the calcium reporter GCaMP6s (*Chen et al., 2013*) in M1 under the control of the glr-2 promoter (*Brockie et al., 2001*) and measured light-induced calcium changes. As indicated by robust GCaMP6s fluorescence increases, light increased M1 somatic calcium levels, consistent with an acute function for M1 in spitting (*Figure 4M–O, Video 15*).

We next asked if lite-1 and gur-3 control M1's calcium response to light. lite-1 is expressed in M1 (*Bhatla and Horvitz, 2015*; *Taylor and Santpere, 2019*). The gur-3 expression pattern was first determined using extrachromosomal arrays expressing GFP under the gur-3 promoter (*Bhatla and Horvitz, 2015*) and subsequently examined using single-cell transcriptomic analysis of the *C. elegans* nervous system (*Taylor and Santpere, 2019*). Although neither method detected gur-3 expression in M1, when we integrated the previously generated extrachromosomal arrays into the genome we noted that, in addition to being expressed in the I2 and I4 neurons as reported previously (*Bhatla and Horvitz, 2015*), the stably integrated gur-3$_{promoter}$::GFP transgene was also expressed in M1 and another pharyngeal neuron, MI (*Figure 3—figure supplement 1M and N*), consistent with possible functions for both lite-1 and gur-3 in M1. As was the case with pharyngeal muscle

calcium increases, both *lite-1* and *gur-3* were required for light-induced calcium increases in the M1 soma (*Figure 4P–U*), with *lite-1*'s playing the major role in the overall response (*Figure 4T*) and *gur-3*'s playing a minor role necessary only in the acute response (*Figure 4U*), suggesting that the differential requirement for *lite-1* and *gur-3* in spitting occurs at or upstream of M1's activation.

We then asked if M1 might be activated directly by light, as suggested by its expression of *lite-1* and *gur-3*, or whether it instead is activated indirectly by input from other light-sensitive neurons via synaptic, humoral (i.e. dense core vesicle), or gap junction inputs. Synaptic and humoral dense-core vesicle signaling respectively require *unc-13*, which encodes the *C. elegans* homolog of UNC13A (*Maruyama and Brenner, 1991*; *Richmond et al., 1999*; *Richmond et al., 2001*), and *unc-31,* which encodes the *C. elegans* homolog of CADPS/CAPS (*Berwin et al., 1998*), so we examined light-induced M1 calcium responses in these mutants. In both *unc-13(s69)* reduction-of-function mutants (*Richmond et al., 1999*)—null mutations are lethal (*Kohn et al., 2000*)—and *unc-31(u280)* presumptive null mutants (*Speese et al., 2007*), M1's calcium response was reduced in magnitude but still significant (*Figure 4—figure supplement 1I–L*). These results suggest that M1 likely can function as a cellular photoreceptor but do not preclude the possibility that M1's activation occurs through residual *unc-13* function, gap junctions, or a pathway in which *unc-13* and *unc-31* function redundantly.

To determine whether acute activation of M1 is sufficient to produce spitting, we expressed channelrhodopsin 2 (ChR2) (*Nagel et al., 2003*; *Boyden et al., 2005*) in M1. Because the light used to activate ChR2 also activates the endogenous pharyngeal response to light, we performed these experiments using *lite-1 gur-3* mutant animals. While no *lite-1 gur-3* mutant animals grown in the absence of the essential ChR2 cofactor all-*trans* retinal (ATR) showed spitting behavior in response to stimulation with 470 nm blue light, 25% of animals grown with ATR did so when illuminated (*Figure 4V and W*). Thus, optogenetic activation of M1 can produce spitting behavior, although not as efficiently as UV light does in the wild-type background.

Next, we asked what neurotransmitter or neurotransmitters M1 uses to activate pharyngeal muscle. M1 expresses the cholinergic marker genes *unc-17* and *cha-1* (*Franks et al., 2009*; *Taylor and Santpere, 2019*), and consistent with a cholinergic function for M1 we found that animals with loss-of-function mutations in genes required for signaling by acetylcholine (ACh) (*unc-17*, *cha-1*) but not by other neurotransmitters (*unc-25*, *eat-4*, *cat-2*, *tdc-1*, *tph-1*, and *ser-4; mod-1 ser-1 ser-7*) were severely defective in burst pumping, as were *ric-1* mutants, which are defective in nicotinic ACh receptor maturation (*Halevi et al., 2002*; *Figure 4—figure supplement 1M–V*).

We assayed the burst pumping behavior of 36 strains, each carrying a mutation in one of 36 of the 37 known *C. elegans* ACh receptors or receptor subunits (null mutation in the 37th receptor subunit, *acr-9*, results in lethality) (*The C. elegans Deletion Mutant Consortium, 2012*). While several strains showed modest defects, none was strikingly defective, suggesting that multiple acetylcholine receptors function redundantly downstream of M1 (*Figure 4—figure supplement 2A–P'*).

Together these observations confirmed and extended our previous finding that M1 is necessary for spitting, additionally showing that M1 controls spatially-restricted muscle calcium signals, that it is acutely activated by light, and that its activation can suffice for spitting.

## The *gur-3*-expressing I2 and I4 neurons can weakly activate M1 to open the metastomal filter and pharyngeal valve without increasing pumping rate

The M1 neuron's expression of *lite-1* and *gur-3* suggests that M1 is a likely site of action of *lite-1* and *gur-3* in the spitting response to light. However, *unc-13* and *unc-31* mutants were significantly reduced in M1 activation (*Figure 4—figure supplement 1I–L*), consistent with the possibility that other *lite-1* and/or *gur-3* neurons provide stimulatory inputs to M1.

To identify light-sensing neurons that function upstream of M1, we first focused on pharyngeal neurons known to express *gur-3*. The pharyngeal I2 neurons express *gur-3* and control light-induced pumping inhibition (*Bhatla and Horvitz, 2015*). Since the I2s form electrical and chemical synapses with M1 (*Albertson and Thomson, 1976*), we tested the hypothesis that they provide functional inputs to M1 for the induction of spitting by expressing a wild-type copy of *gur-3* under an I2-specific promoter (*Bhatla and Horvitz, 2015*) in *lite-1 gur-3* double mutants. I2-specific rescue of *gur-3* restored robust light-induced spitting, opening of the metastomal filter, and weak M1 calcium

increases to *lite-1 gur-3* double mutants, indicating that light can activate M1 via *gur-3* function in the I2 neurons (*Figure 5A–E*).

Laser ablation of M1 suppressed this *gur-3*-dependent spitting and opening of the metastomal filter (*Figure 5F–H*), showing that the I2 neurons act via (or possibly in parallel to) M1 to induce spits and open the filter.

Since the I2 neurons can strongly stimulate spitting, we next asked if they could increase pumping rates in slow-pumping *eat-2* mutants. Light did not increase the pumping rate of *eat-2; lite-1 gur-3; I2_{promoter}::gur-3(+)* animals (*Figure 5I*). Thus, while the I2 neurons can activate M1 and thereby open the pharyngeal valve and metastomal filter, they are unable to activate M1 sufficiently strongly to drive pumping rate increases. This finding is consistent with our calcium imaging results indicating the I2 neurons can activate M1 only weakly (*Figure 5D–E*) and with our observation that *lite-1*, but not *gur-3,* is required for burst pumping (*Figure 3—figure supplement 1H–L*).

We also observed that *lite-1 gur-3; I2_{promoter}::gur-3(+)* animals inhibited pumping only weakly, if at all (*Figure 5B and F*). This finding was initially surprising, because prior work showed that the I2s inhibit pumping in response to light (*Bhatla and Horvitz, 2015*). Interestingly, after we ablated M1 in these animals, we observed robust light-induced pumping inhibition (*Figure 5G and J*). This finding suggests that the I2 neurons' stimulatory effect on M1 outweighs their inhibitory effect on pharyngeal muscle. Initially, it seemed paradoxical that the I2s would function in this manner. However, when we compared the rate of feeding pumps (i.e. pumps that capture food) in M1- and mock-ablated *lite-1 gur-3; I2_{promoter}::gur-3* animals, it became clear that the I2s' activation of M1 serves to reduce the overall feeding rate, because activating M1 eliminates feeding pumps by turning them into spits (*Figure 5K*). Thus, rather than simply acting as nonspecific inhibitors of the rates of both feeding and spitting pumps, the I2s reduce food intake via two mechanisms: direct inhibition of feeding by hyperpolarizing pharyngeal muscles (*Bhatla and Horvitz, 2015*; *Bhatla et al., 2015*) and indirect transformation of feeding pumps into spits by activating M1 (*Figure 5L*).

We used an I2-specific genetic-ablation transgene (*Bhatla and Horvitz, 2015*) to kill the I2 neurons in wild-type and *lite-1* genetic backgrounds. Spitting persisted after ablation of the I2 neurons in both cases (*Figure 6A and B*), confirming that the I2 neurons are not necessary for spitting. Thus, *gur-3*-expressing neurons other than the I2 neurons can promote spitting.

*gur-3* is also expressed in the pharyngeal I4 neuron (*Figure 6C*), which has no known function. I4 is activated by light (*Bhatla and Horvitz, 2015*) and synapses onto M1 (*Albertson and Thomson, 1976*). To determine if I4 contributes to M1's *gur-3*-dependent activation by light, we laser-killed the I2s and I4 of *lite-1* mutants singly and in combination and assayed spitting behavior. No ablation condition completely eliminated spitting (*Figure 6D–H*), but I4 ablation significantly reduced the spitting of *lite-1* I2-ablated animals (*Figures 6E, G and H*), suggesting that I4 promotes spitting in *lite-1* I2-ablated animals. We did not observe significant differences in spitting or pumping inhibition between *lite-1* I4-ablated and mock-ablated animals (*Figures 6D, F, H and I*), so our conclusion that I4 promotes spitting is limited to the context of *lite-1* animals lacking I2 neurons. More generally, the presence of I2s creates two issues that affect the interpretation of all neuronal ablations in *lite-1* animals. First, when the I2 neurons are present they inhibit pumping, and this I2-dependent inhibition compresses the dynamic range of the assay by ~50% (*Figure 6I*). This compression thus has the potential to mask the contributions of non-I2 neurons to spitting and makes it difficult to accurately assay the effect of a given ablation on spitting. Second, when the I2s are present in a *lite-1* background this assay cannot distinguish between an increase in inhibition and a decrease in spitting. These limitations are specific to *lite-1* animals, because *lite-1* animals are so defective in pumping inhibition that the burst pumping phase is not clearly visible. For these reasons, we cannot interpret any ablation performed in the context of functional I2s in a *lite-1* background.

We detected small light-induced M1 calcium increases in mock-, I2-, and I4-ablated *lite-1* animals but not in I2/I4 double-ablated *lite-1* animals (*Figure 6—figure supplement 1A–1C*), and I2-, I4-, and I2/I4-ablated animals seemed to open the metastomal filter slightly less often than did mock-ablated controls (*Figure 6—figure supplement 1D*), but these differences were not statistically significant. We noticed that the pumping inhibition of *lite-1* I4-ablated animals (*Figure 6F*) was much stronger than that of *lite-1 gur-3; I2_{promoter}::gur-3(+)* animals (*Figure 5D and H*), suggesting that *gur-3* might function in neurons other than the I2s and I4 to regulate the balance between promotion and inhibition of pumping. Finally, these results are also consistent with *gur-3*'s functioning in

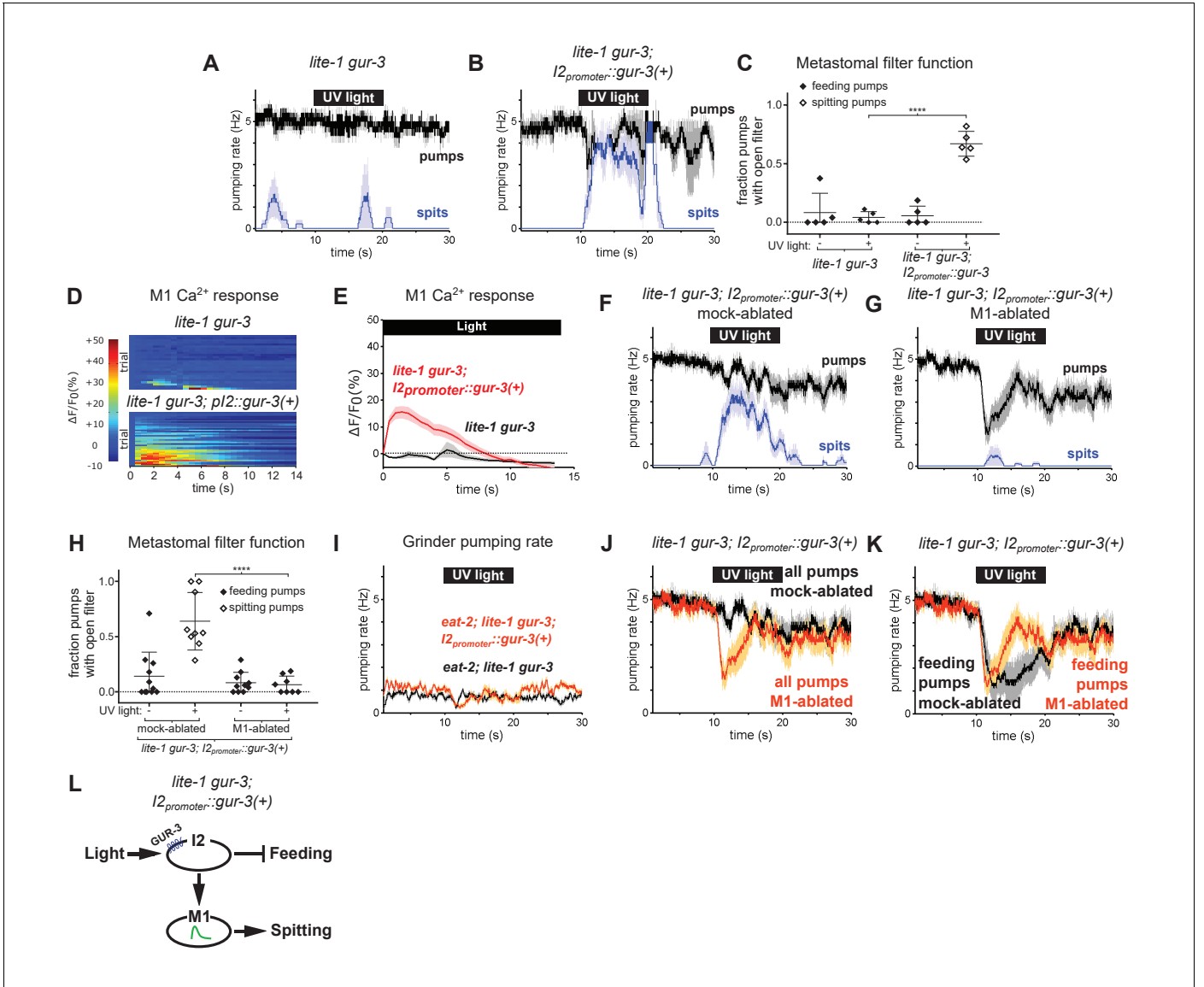

**Figure 5.** The *gur-3*-Expressing I2 Neurons Can Activate M1 to Produce an Attenuated Form of Spitting. (**A–B**) I2-specific expression of *gur-3* restores light-induced spitting to *lite-1 gur-3* mutants. n = five animals. (**C**) Quantification of the frequencies at which the metastomal filter was open in feeding and spitting pumps in (**A**) and (**B**). Center bar, mean; error bars, SEM; ****, p < 0.0001; t test. n = five animals. (**D**) I2-specific expression of *gur-3* restores the light-induced M1 calcium response of *lite-1 gur-3* mutants. M1 somatic GCaMP responses of individual animals to light. Each row is a different animal; ΔF/F$_0$ over time is indicated according to the heatmap at left. n ≥ 26 animals. (**E**) Average of (**D**). (**F–G**) Laser ablation of M1 suppresses the I2-specific *gur-3* rescue of *lite-1 gur-3* spitting defects. n = 11 animals. (**H**) Quantification of the frequencies at which the metastomal filter was open in feeding and spitting pumps in (**F**) and (**G**). Center bar, mean; error bars, SEM; ****, p < 0.0001; t test. n ≥ 8 animals. (**I**) I2-specific expression of *gur-3* via transgene *nls791* (*I2$_{promoter}$::mcherry::gur-3(+)*) does not rescue the burst pumping defect of *eat-2; lite-1 gur-3* animals in the grinder pumping assay. n = 20 animals. (**J**) Comparison of grinder pumping rates in (**F**) and (**G**). While mock-ablated animals from (**F**) do not inhibit pumping in response to light, M1-ablated animals from (**G**) exhibit robust light-induced inhibition, indicating that the I2s' promotion of pumping via M1 outweighs their inhibition of pumping. (**K**) Comparison of feeding pumping rates from (**F**) and (**G**). (**L**) Model for the effects of the I2 neurons on M1 and pumping. The I2 neurons reduce overall feeding by two mechanisms: direct inhibition of feeding pumps and M1-dependent promotion of spitting pumps. Shading around traces indicates SEM. Unless indicated otherwise, 'I2-specific expression of *gur-3*' refers to transgene *nEx2144* (*I2$_{promoter}$::mcherry::gur-3(+)*). 'Genetic ablation of the I2 neurons' refers to transgene *nls569* (*I2$_{promoter}$::csp-1b*) (*Bhatla and Horvitz, 2015*). '*I2$_{promoter}$*' refers to the *flp-15* promoter.

The online version of this article includes the following source data for figure 5:

**Source data 1.** Source data for *Figure 5*.

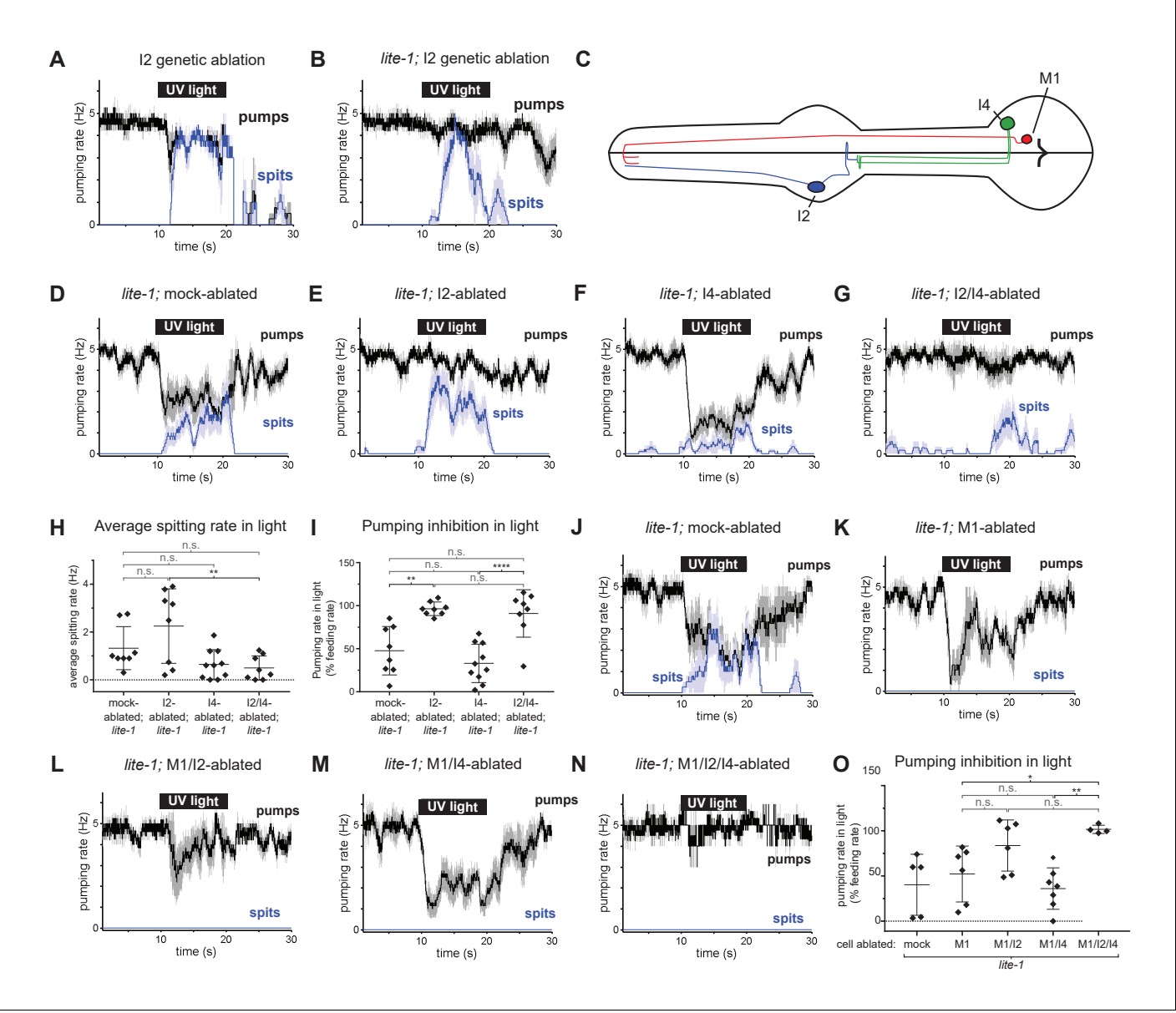

**Figure 6.** The I4 neuron functions with the I2 neurons to promote spitting and inhibit feeding. (**A**) Genetic ablation of the I2 neurons in a wild-type background does not eliminate light-induced spitting. n = five animals. (**B**) Genetic ablation of the I2 neurons of *lite-1(ce314)* mutants does not eliminate light-induced spitting. n = six animals. (**C**) Diagram indicating the cellular morphology of M1 and the *gur-3*-expressing I2 and I4 pharyngeal neurons. The I2 neurons are bilaterally symmetric; only one I2 neuron is depicted. (**D**) Mock-ablated *lite-1* animals inhibit pumping and spit in response to light. n = eight animals. (**E**) I2-ablated *lite-1* animals spit in response to light but do not inhibit pumping. n = eight animals. (**F**) I4-ablated *lite-1* animals inhibit pumping and spit in response to light. n = 10 animals. (**G**) I2/I4 double-ablated *lite-1* animals exhibit delayed spitting and do not inhibit pumping in response to light. n = eight animals. (**H**) Quantification of average spitting rate during light exposure of animals from (**C–D**). **, adjusted p value < 0.005; n.s., not significant; ordinary one-way ANOVA. (**I**) Quantification of pumping inhibition during light exposure of animals from (**D–G**). ****, approximate adjusted p value < 0.0001; **, adjusted p value < 0.005; n.s., not significant; ordinary one-way ANOVA. (**J**) Mock-ablated *lite-1* animals inhibit pumping and spit in response to light. n = five animals. (**K–M**) M1-ablated (n = six animals), M1/I2-ablated (n = 6), and M1/I4-ablated (n = 7) *lite-1* animals inhibit pumping, but do not spit, in response to light. (**N**) M1/I2/I4 triple-ablated *lite-1* animals neither inhibit pumping nor spit in response to light. n = four animals. (**O**) Quantification of pumping inhibition in (**J–N**). Ablation of the I2 neurons eliminates light-induced pumping inhibition by M1/I4-ablated animals. **, adjusted p value < 0.005; *, adjusted p value < 0.05; n.s., not significant; ordinary one-way ANOVA. Shading around traces indicates SEM. All animals were *lite-1*(ce314) null mutants and carried transgene *nIs678 (M1_{promoter}::gcamp6s)*. '*M1_{promoter}*' refers to the *glr-2* promoter. Center bar, mean; error bars, SD.

The online version of this article includes the following source data and figure supplement(s) for figure 6:

**Source data 1.** Source data for *Figure 6*.

*Figure 6 continued on next page*

*Figure 6 continued*

**Figure supplement 1.** Additional characterization of the effects of ablating the I2 and/or I4 neurons in *lite-1* mutant animals.

**Figure supplement 1—source data 1.** Source data for *Figure 6—figure supplement 1*.

another upstream neuron that acts via I4, rather than in I4 itself. In short, our ablation data suggest but do not prove that I4 can promote spitting.

To verify that the spitting we observed in these experiments was M1-dependent, we killed M1 in combination with the I2 and I4 neurons of *lite-1* mutants. As expected, ablation of M1 eliminated spitting in all conditions (*Figure 6J–O*). Consistent with our observation that I2 inhibits pumping in M1-ablated animals (*Figure 5G*), we found that M1- and M1/I4-ablated animals inhibited pumping in response to light, while triple-ablated animals lacking M1, I2, and I4 were defective in pumping inhibition (*Figure 6K and M–O*). Some M1/I2-ablated animals inhibited pumping in response to light, suggesting that the I4 neuron might inhibit pumping in a manner similar to the I2 neurons, but this effect was not statistically significant (*Figure 6L and O*).

Because ablating the I2 and I4 neurons together failed to completely eliminate spitting, *gur-3* likely functions in one or more additional neurons. In addition to being expressed in M1 and MI, *gur-3* is also expressed in the extra-pharyngeal AVD neurons. These cells or others might be additional sites of residual *gur-3* function. In short, the I2 and I4 neurons and perhaps others likely function in parallel to promote *gur-3*-, M1-dependent spitting.

## Discussion

To understand how the nervous system adapts the output of a single set of muscles to a changing environment, we analyzed the spitting reflex of the *C. elegans* pharynx. This reflex comprises three parallel behavioral components: an increase in pumping rate, the opening of the pharyngeal valve at the anterior end of the procorpus, and the opening of the metastomal filter. Opening the pharyngeal valve permits the expulsion of ingested material (i.e. spitting), while opening the metastomal filter allows the worm to flush in and out large amounts of fluid and particles not previously in the pharynx, apparently rinsing its mouth in response to a bad taste (*Figure 7A*). The pm2 and likely the pm1 pharyngeal muscles open the metastomal filter, while a subcellular compartment of the pm3 muscles opens the pharyngeal valve, transforming feeding motions into spitting motions. The pm3s make up the bulk of the anterior pharynx, which pumps rhythmically during spitting. Opening of the pharyngeal valve by the pm3s requires that a subcellular compartment at the anterior end of each pm3 undergoes a sustained uncoupling from the rest of the cell and remains contracted. Strikingly, the bulk of each pm3 muscle cell continues to pump, ejecting ingested material. The subcellular contraction of the pm3s is accompanied by subcellularly localized increases in intracellular calcium at the site of the pharyngeal valve. Spitting behavior, opening of the metastomal filter, and muscle calcium physiology are controlled by the M1 neuron, which directly innervates the pharyngeal muscles that control the filter and valve. The gustatory-receptor orthologs *lite-1* and *gur-3* function together in the control of light-induced spitting: *lite-1* plays the major role and suffices for nearly wild-type responses to light, while *gur-3* plays a smaller role and is sufficient to partially open the valve and filter but not to increase pumping rate. *gur-3* functions in the I2 pharyngeal neurons and likely also in the I4 neuron to evoke this attenuated spitting response via M1.

### Subcellular muscle units driven by neuronally regulated subcellularly localized calcium signaling modulate motor behavior

We identified a likely physiological driver for the subcellular muscle contraction of the pm3s: subcellularly localized calcium transients in the anterior end of the pm3s, in the specific muscle region that undergoes sustained contraction during spitting and from which the pharyngeal valve protrudes. We propose that these calcium transients are the drivers of the subcellular muscle contraction of the pm3s.

Subcellularly localized calcium signaling in muscle has been observed in other systems. Subcellular muscle calcium transients known as 'calcium sparks' result from calcium influx via individual ryanodine receptors (*Cheng et al., 1993*). These localized calcium increases can trigger calcium-induced

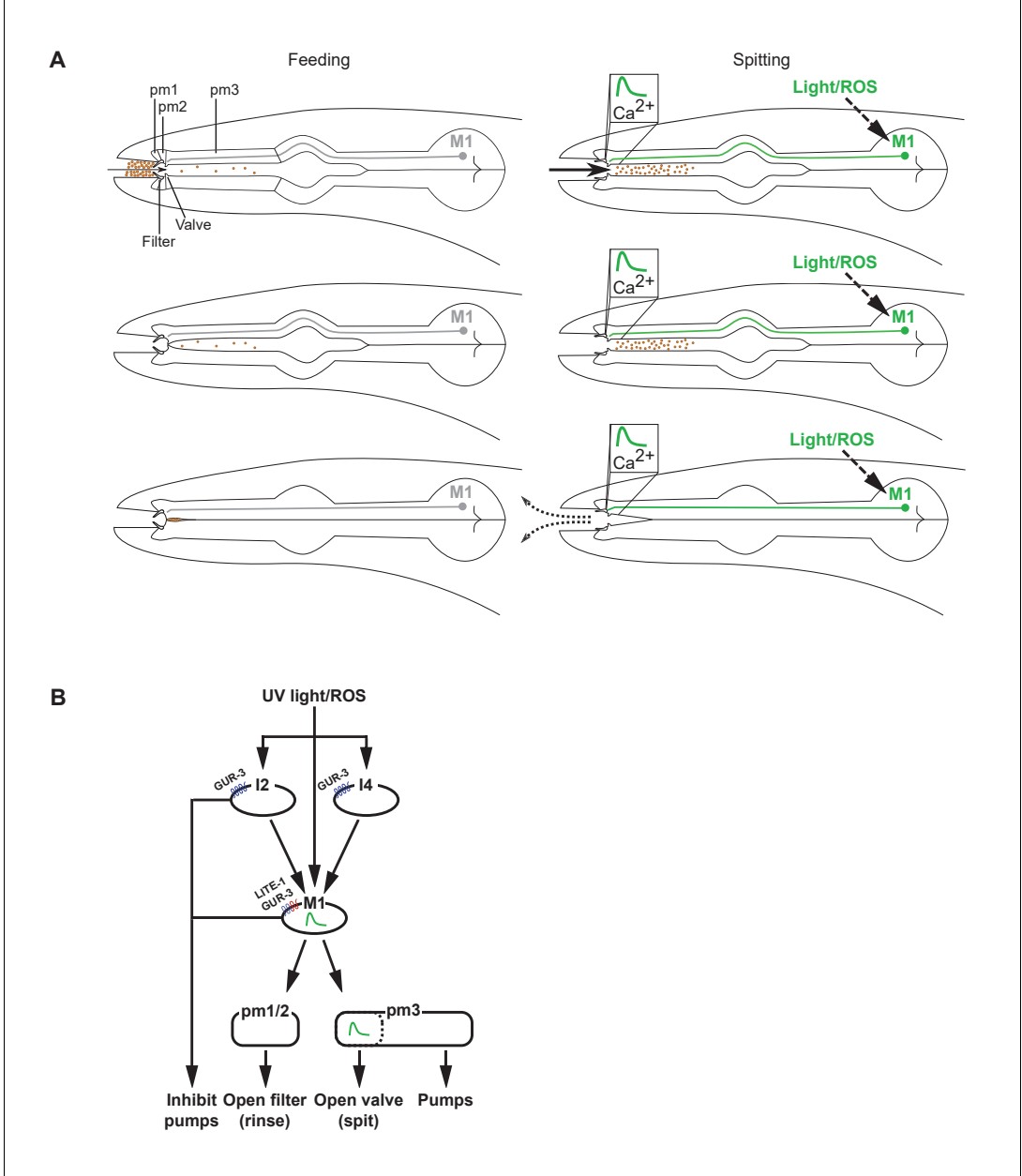

**Figure 7.** Models of spitting behavior and neural circuitry. (**A**) Model for feeding and spitting. During feeding, the M1 neuron is inactive. Thus, the metastomal filter is partially closed, restricting particle influx, and the pharyngeal valve closes at the end of each pump, trapping food. During spitting induced by light or reactive oxygen species (ROS), M1 is active, thus inducing spitting of ingested material by opening the pharyngeal valve and facilitating rinsing of the pharynx by opening the metastomal filter. (**B**) Model for neural circuitry controlling light- and ROS-induced spitting. Light or ROS stimulate the M1, I2, and I4 neurons via LITE-1 and GUR-3. The I2 and I4 neurons excite M1, which is situated in the 'waist' of an 'hourglass' circuit motif. In turn, M1 makes diverging outputs to multiple pharyngeal muscle classes, each driving distinct aspects of the spitting reflex: the pm1 and pm2 muscles open the metastomal filter, a subcellular compartment of the pm3 muscles opens the pharyngeal valve, and the remainder of pm3 pumps. The M1 and I2 neurons each also inhibit pumping.

The online version of this article includes the following source data for figure 7:

**Source data 1.** Source data for *Figure 7*.

calcium release from the sarcoplasmic reticulum, resulting in a wave of calcium release that drives excitation-contraction coupling (*Cheng et al., 1993*; *Kuo and Ehrlich, 2015*). Calcium sparks also allow local increases in intracellular calcium to drive nearby signaling events without elevating global calcium levels (*Nelson et al., 1995*; *Navedo et al., 2005*; *Nieves-Cintrón et al., 2008*).

Despite these examples of subcellularly localized muscle calcium function in physiological settings, how subcellular muscle calcium signaling contributes to behavior is poorly understood. The *C. elegans* pm5 pharyngeal muscle cells function to swallow food from the anterior pharynx to the terminal bulb via a peristaltic wave of subcellular contraction (*Avery and Horvitz, 1987*), and subcellular calcium signals in the pm5s likely underlie swallowing behavior (*Shimozono et al., 2004*; *Kozlova et al., 2019*). These transients have been reported to resemble calcium waves in cardiac muscle (*Kozlova et al., 2019*), a phenomenon driven by the propagation of calcium-induced calcium release from internal stores (*Cheng et al., 1993*). Subcellular signals have also been observed in murine enteric muscles, which can contract peristaltically, but the functional significance of these signals is unknown (*Yamazawa and Iino, 2002*). Our results identify a distinct and novel role for subcellular muscle contraction and calcium signaling: sustained, neuronally induced contraction of a stable, dedicated subcompartment of muscle, thus modulating function and transforming one behavioral output into another. While calcium sparks and *C. elegans* peristalsis behavior occur on a timescale of tens of milliseconds, the contractions and calcium signals that we observed can last at least 10 s (with the *caveat* that our imaging assay lacks the temporal resolution to distinguish a sustained calcium influx from a series of rapid, short-lived calcium pulses). This timescale is more similar to that of calcium sparklets produced by L-type voltage-gated calcium channels (*Navedo et al., 2005*).

We identify the M1 motor neuron, which innervates the anterior region of the pm3 muscles, as the driver of localized pm3 contraction and propose two alternative models for how M1 might produce the spatial and temporal contraction pattern that occurs during spitting. In the first model, the neuromuscular junctions (NMJs) of M1 at the anterior ends of the pm3s drive the entirety of the spitting response—the pm3 subcellular calcium increases and sustained subcellular contraction as well as the rhythmic and global contractions—that is, pumps—of pharyngeal muscle in general. For M1 to evoke sustained contraction of only the anterior ends of the pm3s while also stimulating pumps globally, the contracting region must be functionally distinct from the remainder of pm3. For example, the anterior region of pm3 might express a set of ion channels distinct from that expressed in the posterior region or might be modulated by a locally acting M1-driven signaling cascade that inhibits the relaxation of the anterior subcompartment. In the second model, M1 promotes subcellular contraction and global pumping via distinct sets of NMJs. In this case, the NMJs of M1 at the anterior end of pm3 would open the pharyngeal valve directly, while M1 indirectly stimulates global pumping by exciting one or more other neurons that form NMJs with more posterior regions of the pm3s or with other more posterior pharyngeal muscles. That the muscles of the pharynx are interconnected by gap junctions could allow the stimulation of one muscle to trigger a global contraction of other muscles. Eight pharyngeal neuron classes (I1, I2, I3, I5, M4, M5, MI, NSM), each of which forms NMJs with multiple pharyngeal muscle classes, receive synaptic input from M1 (*Cook et al., 2020*), and at least two of these classes (I1 and M4) can promote pumping (*Trojanowski et al., 2014*) and thus are plausible candidates to drive pan-pharyngeal M1-dependent contractions in response to light. This second model does not require that the anterior tip of pm3 be functionally specialized beyond receiving NMJs from M1, because the threshold for local contraction could be lower than that for depolarizing the entire muscle.

More generally, we suggest that subcellular muscle contraction via subcellular calcium signaling is a mechanism by which the nervous system can modulate muscle function to alter motor outputs and generate distinct behaviors. We suggest that *C. elegans* spitting behavior offers a system for the analysis of the molecular bases of subcellular calcium signaling in muscle using a highly genetically tractable organism.

## Independent control of the pharyngeal valve and metastomal filter generates four distinct pumping behaviors

We observed that, in addition to opening the pharyngeal valve, spitting worms open the metastomal filter and this opening depends in part on the pm2 muscles. This finding suggests that in addition to passively filtering incoming material as reported previously (*Fang-Yen et al., 2009*), the metastomal filter can also be actively opened to allow unrestricted entry of material into the pharynx. In the context of spitting pumps, opening the filter results in new material being flushed in and out, while closing the filter results in egestion without significant admission of new particles. We speculate that, in combination with the opening of the pharyngeal valve, this opening of the metastomal filter produces a rinsing effect, in which greatly increased quantities of material are drawn into and then

expelled from the pharynx. Such a mechanism might act to eliminate toxic material before it is ingested as the animal—in response to an aversive stimulus—moves away from the source of such noxious chemicals into a less toxic neighborhood.

We observed that the filter sometimes also appears to open during feeding, producing 'gulps' with greatly increased intake. Thus, the independent control of the metastomal filter and pharyngeal valve produces four distinct pumping behavior variants—feeding, gulping, spitting with rinsing, and spitting without rinsing. Given that M1 is the major source of innervation of the pm1 and pm2 muscles, from which the metastomal filter protrudes (*Albertson and Thomson, 1976*), we think it likely that M1 can function in the production of gulps by opening the metastomal filter without producing an accompanying contraction in the regions of pm3 that open the pharyngeal valve. Such a behavior might allow enhanced consumption of certain preferred food or, because it draws large amounts of material into the mouth, might allow the worm to more efficiently sample the environment while foraging. The possibility that a single neuron might control opposite behaviors (i.e. spitting and gulping) via the differential activation of multiple downstream muscles and subcellular muscle compartments is intriguing.

## The M1 neuron drives spitting from the waist of an hourglass circuit motif

Our circuit studies show that the M1 motor neuron is a key regulator of spitting. We speculate that in addition to controlling the muscles that produce spitting, M1 functions as a central integrator of noxious taste information in the pharyngeal nervous system. Several lines of evidence support this hypothesis. First, a recent analysis of the pharyngeal connectome found that M1 exhibits a prominent and highly connected position in the pharyngeal network (*Cook et al., 2020*), suggesting that M1 plays an important role in pharyngeal network function. Second, M1 is particularly well-positioned to integrate noxious taste stimuli, as electron microscopy-based reconstructions of the pharyngeal nervous system indicate that M1 receives direct synaptic inputs from each of the other five *lite-1*- and/or *gur-3*-expressing pharyngeal neurons (i.e. the paired I2 and single I4, M4, M5, and MI neurons) (*Albertson and Thomson, 1976*; *Bhatla and Horvitz, 2015*). Consistent with this reported connectivity, we found that two of these neuron classes, the I2s and I4, contribute to M1's activation by light. Third, it is likely that M1 is itself light- and ROS-sensitive, as it expresses both *lite-1* (*Bhatla and Horvitz, 2015*) and *gur-3* (this study) and we show here that it can respond robustly to light even when synaptic and humoral inputs are substantially reduced or eliminated, respectively. For these reasons, we propose that M1 sits at the narrow waist of an 'hourglass' circuit motif (*Figure 7B*). In this motif, multiple upstream parallel pathways converge on M1, which in turn makes divergent outputs onto the multiple downstream muscle groups classes that control the discrete motor outputs of the spitting reflex. Such an hourglass structure was recently described in a theoretical analysis of the larger *C. elegans* non-pharyngeal nervous system (*Sabrin et al., 2020*). We propose that the neuromuscular circuit we describe here constitutes a miniature variation of such an hourglass circuit organization.

## The magnitude of M1 activation by the I2 and I4 neurons evokes a distinct subset of hourglass motor outputs

We explored circuitry upstream of the M1 neuron and identified the I2 and I4 neurons as likely sites of *gur-3* function in light-induced spitting. The I2s are activated by and inhibit pumping in response to light, and I4 was previously shown to be activated by light (*Bhatla and Horvitz, 2015*), but no function for I4 had previously been reported. Because I4 expresses *gur-3* and controls responses to light, we speculate that I4 is also a cellular receptor for light and ROS. Interestingly, although the pharyngeal connectome and our data suggest that M1 acts as a bottleneck through which signals must pass to evoke spitting, the I2/I4 circuit can selectively produce a subset of M1-dependent behaviors by weakly activating M1. Weak *gur-3*-dependent stimulation of M1 is sufficient to open the pharyngeal valve and metastomal filter, but not to increase pumping rates. Because UV light robustly activates the I2s (*Bhatla and Horvitz, 2015*), the fact that the I2s activate M1 only weakly suggests that their input to M1 is similarly weak. The weakness of this connection might be a design feature of the pharyngeal nervous system, allowing strong activation of the I2 neurons to be converted into weak activation of the M1 neuron, thereby producing an attenuated variation of spitting.

We speculate that I4 might function in a similar way, and that the I2s and I4 serve as modules to trigger attenuated spitting, while input from other cells and/or endogenous *lite-1* function in M1 produce spitting in a more robust manner, enabling the pharyngeal nervous system to generate graded variations of the spitting reflex and thus fine-tune its output to the environment. One problem that must be solved by an hourglass circuit structure with one or a few cells in the bottleneck is how to differentially evoke downstream outputs when all upstream inputs must be compressed together at the hourglass waist. We suggest that the M1 spitting circuit illustrates a simple solution to this problem: graded activation of M1 in the hourglass waist activates progressively larger subsets of downstream outputs. Outputs of such a circuit can thus be tuned by adjusting the magnitude of upstream inputs, allowing upstream neurons to input additively if their connections to the bottleneck are equally weighted or to evoke a subset of downstream outputs if they only weakly activate the bottleneck (as is the case with the I2 neurons).

Previous studies showed that the I2s inhibit pumping and that M1 is required for spitting (*Bhatla and Horvitz, 2015*; *Bhatla et al., 2015*). We found that in addition the I2s can promote pumps (spitting pumps) and that M1 can inhibit pumping (it does so in the recovery phase). Thus, each of these neuron classes can function in superficially opposite ways in different contexts. Although it might seem counterproductive for a given neuron to both promote and inhibit pumping, both spitting and the inhibition of pumping have the common effect of reducing food intake, since spitting pumps eject previously ingested food. We suggest that these differentially timed inhibitory and excitatory neural inputs to pharyngeal muscle allow the pharynx to generate a sequence of actions, first inhibiting pumping and/or stimulating spitting to eject noxious material, then increasing pumping rate and rinsing in the burst pumping phase if noxious stimulation continues, and finally inhibiting all pumping, including spitting, as the animal flees the site of the noxious encounter. Additionally, because spitting with an opened metastomal filter results in rinsing new material into the pharynx, it might be maladaptive if spitting persists when the animal is in a large region of noxious material; in this case, circuits promoting inhibition on a longer timescale than spitting would serve to limit additional exposure to noxious material.

In conclusion, we propose that subcellular muscle contraction induced by subcellular calcium signaling is a fundamental mechanism by which nervous systems can reshape motor behavior and suggest that hourglass motifs might function broadly in neuronal circuit organization for the production of complex adaptive behavior in response to changing environments.

## Materials and methods

### Molecular biology

Transgenes were generated by standard cloning methods. DNA sequences used in the generation of transgenes were amplified with the primers indicated below.

Generated using the infusion cloning technique (Takara Bio, Mountain View, CA) and injected as plasmids:

nIs864 - pSS028 [glr-2$_{promoter}$::gfp::unc-54 3' UTR]

nEx2905 and nIs865 - pSS042 [lury-1$_{promoter}$::ice::sl2::mcherry::unc-54 3' UTR]

nEx2815 - pSS034 [lury-1$_{promoter}$::chr2::sl2::mcherry::unc-54 3' UTR].

glr-2$_{promoter}$: TTGGGACAAATGTGGAAACGA, TTCGCTTTTTACAGAGTTAACTCTGC

lury-1$_{promoter}$: TTCATTAAGTAATCCGTTTAGGGCAAAT, GATTGGATTTTCTGGAATAATCGGGT

ice: ATGGCCGACAAGGTCCTGA, TTAATGTCCTGGGAAGAGGTAGAAACATC,

Generated and injected as PCR product:

nIs686 - pcNB1 [gpa-16$_{promoter}$::gcamp3::unc-54 3' UTR]

nEx2157 – pcNB20 [gur-3$_{promoter}$::gur-3 gDNA::gur-3 3' UTR]

nIs678 - [glr-2$_{promoter}$::gcamp6s::unc-54 3' UTR],

gpa-16$_{promoter}$: ACCAACCTGAACAGCGAATC, AAAGGGAATTTTATGTGAATAATATCG

gur-3$_{promoter}$::gur-3 gDNA::gur-3 3' UTR: GCCTGATGGAACACACTTCCAAC, GTTTACCCCGTCTCTATTTCCGCTTTG

gcamp6s: ATGGGTTCTCATCATCATCATCATC, GCCCGTACGGCCGACTAGTAGG.

## Grinder pumping assay

Previously, we used a xenon bulb (Till Photonics Polychrome V, 150 W; Thermo Fischer Scientific, Waltham, MA) to deliver 365 nm UV light through an inverted microscope (Axiovert S100; Zeiss, Oberkochen, Germany) to individual worms on an agar pad (*Bhatla and Horvitz, 2015*). Because of the low throughput of this approach, we designed a device to deliver 365 nm UV light to standard Petri dishes used for culturing worms and used this apparatus to assay behavioral responses to 365 nm UV light as described previously for 436 nm violet light (*Bhatla and Horvitz, 2015*). The apparatus consists of a UV-light-emitting LED (M365F1; Thorlabs, Newton, NJ) coupled by a fiber-optic cable to a collimating lens. The LED system delivers 365 nm UV light in a 3–4 mm spot at a power of ~1.5 mW/mm$^2$ to young (1 day post-L4) hermaphrodite adults moving freely on a Petri dish, producing a burst pumping response indistinguishable from that obtained previously (*Bhatla and Horvitz, 2015*). Custom Matlab software (available in *Source code 1*) was used to control the LED, and the contraction rate of the grinder before, during, and after application of light was observed by eye via a dissecting microscope and manually recorded via this software (*Bhatla and Horvitz, 2015*). The data in *Figure 1B*, *Figure 5I*, *Figure 3—figure supplement 1A–L*, *Figure 4—figure supplement 1B, D and M–V*, and *Figure 4—figure supplement 2A–P'* were collected using this method. Genotypes were not typically scored by a researcher blinded to the conditions, but in many cases, we performed both real-time scoring and automated video recording of identical genotypes and always observed similar burst pumping behavior in both cases. Specifically, *Figure 1B* is reproduced by *Figure 1D and E*, *Figure 3—figure supplement 1C, F and G* is reproduced by *Figure 3A–G*, and *Figure 4—figure supplement 1B* is reproduced by *Figure 4—figure supplement 1G*. All behavioral data were visualized as described previously (*Bhatla and Horvitz, 2015*; *Bhatla et al., 2015*). All behavioral assays and imaging experiments were performed in the same location, at which temperature averaged 19°C, with a standard deviation of 1.2°C.

## Pharyngeal transport assay

Pharyngeal transport assays were performed as described previously (*Bhatla et al., 2015*). To visualize the direction of particle motion in the pharynx during each pump, worms were fed a 50% dilution of a 2.5% wt by volume (4.55 x 10$^{10}$ particles per mL) solution of 1 μm diameter polystyrene beads (Polysciences, Inc Warrington, PA, Catalog number 07310–15) in M9 buffer. Young (1 day post-L4) hermaphrodite adults were placed with bacteria on an NGM agar pad on a coverslip and observed with a 20x objective on an inverted microscope (Axiovert S100; Zeiss). Videos were recorded at 1000 frames per sec using a high-speed video camera (Fastcam Mini; Photron, Tokyo, Japan). 365 nm UV light (0.88–1.5 mW/mm$^2$) was presented to the worm for 10 s (Till Photonics Polychrome V). Unlike the burst pumping assay described above, which illuminates the entire body of the worm, this method selectively illuminates the animal's head. Hydrogen peroxide was administered as described previously (*Bhatla and Horvitz, 2015*). Videos were viewed and manually annotated as reported previously (*Bhatla et al., 2015*), using custom Matlab software (available in *Source code 1*), for specific behavioral events: ingestion (beads being retained in the procorpus after procorpus relaxation), spitting (beads being pushed out of the procorpus), and pumping (movement of the grinder). In some cases, the worm moved temporarily out of the focal plane of the microscope or field-of-view of the camera. These unscorable frames were annotated as 'missing' and excluded from further analysis. We were unable to resolve the flaps of the metastomal filter directly, and the position of these flaps was inferred based on whether beads passed freely through or accumulated in the buccal cavity during feeding or spitting, and the fraction of feeding and spitting pumps with the filter open or closed was calculated for the period before and during illumination. Animals with fewer than five scorable pumps in a given 10 s interval were censored from these analyses. Videos were not scored with experimenters blinded to genotype and/or condition.

## Muscle motion visualization assay

To visualize the timing of pharyngeal muscle motions during each pump, young (1 day post-L4) hermaphrodite adults were placed with bacteria on an NGM agar pad underneath a coverslip and observed with a 20x objective on an inverted microscope (Axiovert S100; Zeiss). Videos were recorded at 86 frames per sec using an EMCCD camera (iXon$^+$; Andor, Belfast, United Kingdom). 365 nm UV light was presented to the worm for 10 s (Till Photonics Polychrome V). Slow-moving

*unc-29(e1072)* animals were filmed to facilitate video capture. Videos were viewed and manually annotated based on whether the anterior tip of the pharynx closed or remained open at the end of each pump.

## Laser ablations

We used a pulsed nitrogen laser (MicroPoint; Andor) coupled to a compound microscope (Axioplan; Zeiss) to conduct laser microsurgery of individual pharyngeal neurons, as previously described (*Fang-Yen et al., 2012*). Briefly, worms were immobilized using 10 mM sodium azide and mounted for viewing through a 100x oil objective. Ablations were performed on hermaphrodite larvae of stages 1 or 2 (L1 or L2), with cell nuclei identified by either Nomarski differential interference contrast optics or cell-specific GFP reporters. Specifically, M1 was identified by Nomarski optics or by the M1-expressed GFP reporter *nIs864*, which drives expression of GFP in M1 via the *glr-2* promoter (*Brockie et al., 2001*), while pm1 was identified by GFP reporter *nIs507*, which drives expression of GFP in pm1 via the *inx-4* promoter (*Altun et al., 2009*). The pm2 muscles, the I2 and I4 neurons, and, in some experiments, the M1 neuron were identified by Nomarski optics. Ablations were confirmed the following day by appearance under Nomarski optics and/or by the loss of cell-specific fluorescence. On day 3 or 4, adults were assayed for their response to light. Mock-ablated animals were mounted alongside ablation animals and received identical treatments (i.e. observation of nuclear morphology and/or GFP expression) at each step. Animals were allocated into mock and ablation groups randomly.

## Calcium imaging

Calcium imaging was performed as described previously (*Bhatla and Horvitz, 2015*). Calcium changes in pharyngeal muscle were monitored using GCaMP3 (*Tian et al., 2009*) expressed from one of three transgenes: *nIs686* (this study), driven by the *gpa-16* promoter (*Jansen et al., 1999*); *cuIs36* (*Kozlova et al., 2019*), driven by the *myo-2* promoter (*Okkema and Fire, 1994*; *Okkema et al., 1993*); or *nEx3045* (this study), driven by the *C32F10.8* promoter (*McKay et al., 2003*; *Hunt-Newbury et al., 2007*; this study). Calcium changes in M1 were monitored using GCaMP6s (*Chen et al., 2013*) expressed in M1 using the transgene *nIs678* driven by the *glr-2* promoter (*Brockie et al., 2001*). The free-moving spitting behavior of *nIs686* and *nIs678* animals was indistinguishable from that of wild-type animals (*Figure 2—figure supplement 1H* and *Figure 4— figure supplement 1W*).

   As described by *Kim et al., 2013a*, adult hermaphrodite worms were immobilized on 10% agarose pads under a coverslip using the friction produced by 0.10 µm diameter polystyrene beads (Polysciences, Inc, Catalog number 00876) and imaged using an inverted microscope (Axiovert S100; Zeiss) with a 40x air objective. To stimulate animals with UV light and image GCaMP fluorescence simultaneously, light was flickered between 365 nm UV and 485 nm light at a rate of 2 Hz. Monochromatic 365 nm light, 365 nm/485 nm flickered light, and monochromatic 485 nm light all stimulated immobilized animals to pump, with 365 nm monochromatic and 365 nm/485 nm flickered light evoking pumps more potently than 485 nm light by some strains (*Figure 2—figure supplement 1C and D*). While the response of *nIs686*-bearing animals was indistinguishable from that of the wild type, the response of *nEx3045*-bearing animals was significantly weaker, and *cuIs36*-bearing animals did not pump at all during illumination. This reduced responsivity of some muscle GCaMP3-expressing strains could be a consequence of GCaMP-mediated intracellular calcium-buffering (e.g. as noted by *Singh et al., 2018*) or another abnormality induced by transgene overexpression. To determine whether the light-induced pumps of immobilized animals are spits we illuminated worms immobilized with droplets of mineral oil in their anterior pharynges and filmed the direction in which droplets were drive by pharyngeal pumps. Videos were recorded using an EMCCD camera (iXon[+]; Andor) at two frames per sec and an exposure time of 250 ms, with the exception of videos of animals expressing the *myo-2p::GFP* transgene *vkEx1241*, which were collected with an exposure time of 15 ms, or of animals expressing the *gpa-16p::gcamp3* transgene *nIs686* induced to pump with serotonin, which were collected at four frames per second with an exposure time of 250 ms. Videos were analyzed using custom Matlab software (available in *Source code 1*) as described previously (*Bhatla and Horvitz, 2015*; *Bhatla et al., 2015*). The immobilized pumping assay described in *Figure 2—figure supplement 1C and D* was performed under the same conditions described above

for calcium imaging. Animals that were imaged pumping in response to serotonin were prepared for calcium imaging as described above, with the exception that the M9 solution used to make the agarose imaging pads was supplemented with serotonin hydrochloride (Sigma) to a final concentration of 10 mM.

## Confocal microscopy

Expression patterns of reporter transgenes *nIs686 (gpa-16p::GCaMP3)*, *cuIs36 (myo-2p::GCaMP3)*, *sIs11111 (C32F10.8p::GFP)*, and *nEx3045 (C32F10.8p::GCaMP3)* were determined using a 63x objective lens (Zeiss) on an LSM 800 confocal microscope system (Zeiss). To identify individual *nEx3045* transgene-positive cells responding to light, we first recorded videos of light-induced changes in GCaMP fluorescence as described above, preselecting animals with sparse transgene expression patterns in the anterior pharynx for imaging. We then anesthetized each animal using 50 mM sodium azide and recorded its individual *C32F10.8p::GCaMP3* expression patterns by collecting 20 confocal micrographs evenly spaced across the pharynx. We then used the 'ortho' function of the Zen imaging software (Zeiss) to assemble these micrographs into a 3D reconstruction of GCaMP3 expression. Cells in calcium imaging videos were identified based on their dimensions, unique morphological features (i.e. the pm3s are each characterized by the presence of a branch of the pharyngeal nerve cord, which is visible as a deep, GCaMP-negative groove that runs anterior-to-posterior along the outside surface of each pm3, while the mc1s are characterized by distinctive finger-like projections into the pm3s and by and a sickle-shaped posterior projection into the anterior bulb), and their position relative to other transgene-expressing cells in Z-stack-derived 3D reconstructions of transgenic pharynges.

## Genetic ablation

M1 was ablated genetically by overexpressing the mammalian caspase ICE (*Cerretti et al., 1992*; *Thornberry et al., 1992*; *Zheng et al., 1999*) fused to an *sl2::mcherry* sequence in M1 driven by the *lury-1* promoter (*Ohno et al., 2017*). For all animals assayed, genetic ablation of M1 was confirmed beforehand using an M1-specific GFP marker (*nIs864*), which drives expression of GFP in M1 via the *glr-2* promoter (*Brockie et al., 2001*). The *nEx2905* genetic ablation transgene also killed the M2 neurons in approximately 50% of animals assayed (*Figure 4—figure supplement 1A*), as confirmed using an M2-specific GFP marker (*nIs310*) (*Luo and Horvitz, 2017*). The *sl2::mcherry* sequence used drove varying degrees of mCherry expression in pharyngeal muscle, and we found that the brightness of this mCherry expression correlated with the rate of M2-killing, such that selecting animals with dim pharyngeal mCherry expression reduced the rate of M2-killing to 1–2% (*Figure 4—figure supplement 1C*). Animals assayed in *Figure 4D and E* were not selected in this way. We did not determine the rate of M1 or M2 killing in *nIs865* animals.

## Neuronal silencing via histamine-gated chloride channels

Animals expressing the *Drosophila melanogaster HisCl1* histamine-gated chloride channel (*Pokala et al., 2014*) in M1 under the control of the *glr-2* promoter (*Brockie et al., 2001*) from the transgene *nEx2917 (glr-2p::hiscl1::sl2::mcherry)* were transferred to plates containing 10 mM histamine dihydrochloride (Sigma) and assayed over a 3 hr interval starting 1 hr after exposure to histamine. As controls, transgene-negative animals were tested in the presence of histamine and transgene-positive animals were tested in the absence of histamine. Array-positive animals were selected based on HisCl1::mCherry expression in M1, visualized using a stereofluorescent dissecting microscope (SMZ18, Nikon, Tokyo, Japan). To control for variations in the duration of histamine exposure, array-positive and array-negative animals were tested in alternation.

## Optogenetic depolarization

Optogenetic manipulations were performed as described previously (*Bhatla et al., 2015*). Cultivation plates were seeded with 300 µL of OP50 *E. coli* mixed with 0.5 µL of 100 mM all-*trans* retinal (ATR+) (Sigma) or ethanol alone (ATR-) and stored in the dark. Transgenic hermaphrodite worms carrying the channelrhodopsin 2 (*chr2*) coding sequence (*Nagel et al., 2003*; *Boyden et al., 2005*) fused to a *sl2::mcherry* sequence were cultivated on these plates in the dark from egg to adulthood. For behavioral assays, 1-day-old animals were analyzed as described above ('Pharyngeal Transport

Assay'), except that 470 nm light was used in the place of 365 nm UV light. *nEx2815 [lury-1$_{promoter}$:: chr2::sl2::mcherry::unc-54 3' UTR]* was expressed in both M1 and the M2s driven by the *lury-1* promoter (*Ohno et al., 2017*). To avoid confounding optogenetic activation of the M2s, we assayed only mosaic animals pre-selected for expression of mCherry in M1 but not the M2s. Animals were allocated into ATR- and ATR+ groups randomly.

### Behavioral statistics

For all measurements, each animal was scored once. Biological replicates were performed using separate populations of animals on different days and consisted of 20 animals unless indicated otherwise. Statistics for significance of differences between fractions in metastomal filter analyses were calculated as follows. Fractions were log-transformed, and any fractions with a numerator of zero were approximated to 0.1 prior to transformation. Similar results were obtained by approximating numerators of zero with 0.01. Log-transformed datasets were assessed for significance with an unpaired t-test when a single pair of conditions was compared or unpaired ordinary one-way ANOVA analysis when multiple conditions were compared. These analyses assumed that log-transformed data followed a normal distribution with similar standard deviations. Statistics for significance between average spitting rates were calculated as follows. For each animal, the total number of spits observed during stimulation with light was divided by the number of scorable frames in the light administration phase for that animal to give the average pumping rate during light administration. Significance of differences between groups was assessed by unpaired, ordinary one-way ANOVA analysis, with the assumption that the data followed a Gaussian distribution with similar standard deviations. In the cases in which data did not follow a Gaussian distribution, significance was assessed using the Kruskal-Wallis test, followed by Dunn's multiple comparison test. Pumping inhibition was calculated by dividing the average pumping rate during light stimulation ($10 < t < 20$) by the average pumping rate in the feeding phase ($0 < t < 10$). All statistics were calculated using Prism (GraphPad Software, San Diego, CA).

## Acknowledgements

We thank N An, R Droste, S Flavell, N Ji, E Jorgensen, R Komuniecki, J Meisel, S Mitani, M Treinin, Addgene, and the *Caenorhabditis* Genetics Center, which is funded by NIH Office of Research Infrastructure Programs (P40 OD010440), for strains and reagents; and K Boulias, V Dwivedi, K Burkhart, D Ghosh, H Johnsen, T Littleton, J Meisel, C Pender, P Reddien, J Saul, and Horvitz laboratory members for discussions and advice.

## Additional information

### Funding

| Funder | Grant reference number | Author |
|---|---|---|
| National Institutes of Health | T32GM007287 | Steven R Sando |
| National Institutes of Health | GM024663 | Steven R Sando<br>Nikhil Bhatla<br>H Robert Horvitz |
| Massachusetts Institute of Technology | Friends of the McGovern Institute Fellowship | Steven R Sando<br>Eugene LQ Lee |
| Lord Foundation | Lord Foundation Fellowship | Steven R Sando |
| National Science Foundation | Graduate Research Fellowship | Nikhil Bhatla |
| Agency for Science, Technology and Research | National Science Scholarship | Eugene L Q Lee |
| Howard Hughes Medical Institute | | H Robert Horvitz |
| Miller Institute | Miller Institute Research Fellowship | Nikhil Bhatla |

The funders had no role in study design, data collection and interpretation, or the decision to submit the work for publication.

### Author contributions
Steven R Sando, Conceptualization, Resources, Formal analysis, Validation, Investigation, Visualization, Methodology, Writing - original draft, Writing - review and editing; Nikhil Bhatla, Conceptualization, Resources, Formal analysis, Investigation, Methodology, Writing - review and editing; Eugene LQ Lee, Formal analysis, Writing - review and editing; H Robert Horvitz, Conceptualization, Supervision, Funding acquisition, Writing - review and editing

### Author ORCIDs
Steven R Sando (iD) https://orcid.org/0000-0002-1101-9810
Eugene LQ Lee (iD) https://orcid.org/0000-0003-4725-4959
H Robert Horvitz (iD) https://orcid.org/0000-0002-9964-9613

### Decision letter and Author response
Decision letter https://doi.org/10.7554/eLife.59341.sa1
Author response https://doi.org/10.7554/eLife.59341.sa2

## Additional files
### Supplementary files
• Source data 1. Sequences for plasmids generated in this study.
• Source code 1. Matlab scripts.
• Transparent reporting form

### Data availability
All numerical data and analyses generated during this study are included in the manuscript and supporting files. Each figure and figure supplement is accompanied by a source data file that includes all numerical data used to generate that figure. This includes all excel files, all matlab data files and figures, all statistical analyses, and the. svg (scalable vector graphic) file used to generate each figure. All custom Matlab scripts used in data analysis and the sequences of all plasmids generated in this study are also included in separate source data files. In addition, all raw imaging data (i.e., confocal micrographs, calcium imaging videos, and high-speed behavioral videos) are available for download on figshare (https://doi.org/10.6084/m9.figshare.c.5521161.v1).

The following dataset was generated:

| Author(s) | Year | Dataset title | Dataset URL | Database and Identifier |
|---|---|---|---|---|
| Sando SR, Bhatla N, Lee ELQ, Horvitz HR | 2021 | Source data for Sando et al (2021), eLife | https://doi.org/10.6084/m9.figshare.c.5521161.v1 | figshare, 10.6084/m9.figshare.c.5521161.v1 |

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

# Appendix 1

## Key Resources Table

All strains generated in this study are available upon request from the Horvitz laboratory.

**Appendix 1—key resources table**

| Reagent type (species) or resource | Designation | Source or reference | Identifiers | Additional information |
|---|---|---|---|---|
| Strain, strain background (*Caenorhabditis elegans*) | N2 (wild type) | Horvitz lab collection | WBStrain00000001 | Laboratory reference strain *Figure 1B–G*; *Figure 1—figure supplement 1A*; *Figure 2—figure supplement 1C and D*; *Figures 3A, G and H*; *Videos 1, 2* and *5* |
| Strain, strain background (*C. elegans*) | CB1072 | Horvitz lab collection | WBStrain00004240 | *unc-29(e1072)* *Figure 1—figure supplement 1B and C*; *Videos 3* and *4* |
| Strain, strain background (*C. elegans*) | MT26279 | this study | n/a | *lin-15(n765ts); nIs507 [inx-4$_{promoter}$::gfp; lin-15(+)]* 8x outcrossed to MT8189 *Figure 2B–E*; *Figure 2—figure supplement 1A and B*; *Videos 6, 7* and *8,* and *9* |
| Strain, strain background (*C. elegans*) | MT8189 | Horvitz lab collection | WBStrain00027314 | *lin-15(n765ts)* Used to outcross MT26279, MT23338, and MT24110 |
| Strain, strain background (*C. elegans*) | MT23338 | this study | n/a | *nIs686 [gpa-16$_{promoter}$::gcamp3:: unc-54 3' UTR; lin-15(+)] III; lin-15AB(n765ts)* 5x outcrossed to MT8189 *Figure 2H–J*; *Figure 2—figure supplement 1C, D and H*; *Figure 3I–N*; *Figure 4F and G*, *Videos 10* and *11* |
| Strain, strain background (*C. elegans*) | MT26375 | this study | n/a | *lin-15AB(n765ts); nEx3045 [C32F10.8p:: GCaMP3; lin-15(+)]* *Figure 2—figure supplement 1C, D and G*; *Figure 2—figure supplement 1D*; *Figure 2M and N*; *Video 14* |
| Strain, strain background (*C. elegans*) | MT25732 | this study | n/a | *lin-15(n765ts); nIs864; nEx2905 [lury-1$_{promoter}$::ice:: sl2::mcherry::unc-54 3' UTR; ttx-3$_{promoter}$::mcherry]* *Figure 1—figure supplement 1D*; *Figure 4D and E*; *Figure 4—figure supplement 1B, G and H* |
| Strain, strain background (*C. elegans*) | OK1020 | *Kozlova et al., 2019* | n/a | *cuIs36 X* *Figure 2—figure supplement 1D*; *Figure 2K and L*; *Video 13* |
| Strain, strain background (*C. elegans*) | MT23370 | this study | n/a | *nIs686 III; lite-1(ce314) gur-3(ok2245) lin-15(n765ts)* *Figure 2—figure supplement 1E*; *Figure 3I and L–N*; *Video 12* |

*Continued on next page*

*Appendix 1—key resources table continued*

| Reagent type (species) or resource | Designation | Source or reference | Identifiers | Additional information |
|---|---|---|---|---|
| Strain, strain background (*C. elegans*) | VK1241 | CGC | WBStrain00040040 | *vkEx1241 [nhx-2p::mCherry ::lgg-1 + myo-2p::gfp]* **Figure 2—figure supplement 1F** |
| Strain, strain background (*C. elegans*) | MT26333 | CGC (not outcrossed); this study (outcrossed) | WBStrain00002254 (not outcrossed); n/a (outcrossed) | *sIs11111 [rCesC32F10.8::gfp + pCeh361]* 8x outcrossed to N2 **Figure 2—figure supplement 2C** |
| Strain, strain background (*C. elegans*) | MT2249 | Horvitz laboratory collection | n/a | *egl-47(n1082dm)* **Figure 3—figure supplement 1A** |
| Strain, strain background (*C. elegans*) | RB850 | CGC | WBStrain00031563 | *egl-47(ok677)* **Figure 3—figure supplement 1B** |
| Strain, strain background (*C. elegans*) | MT21783 | *Bhatla and Horvitz, 2015* | n/a | *gur-3(ok2245)* **Figure 3—figure supplement 1C; Figure 3B and G and H** |
| Strain, strain background (*C. elegans*) | RB845 | CGC | WBStrain00031558 | *gur-4(ok672)* **Figure 3—figure supplement 1D** |
| Strain, strain background (*C. elegans*) | FX06169 | Dr. S. Mitani/ NBRP | WBVar01474061 (*tm6169*) | *gur-5(tm6169)* **Figure 3—figure supplement 1E** |
| Strain, strain background (*C. elegans*) | KG1180 | CGC | WBStrain00023485 | *lite-1(ce314)* 5x outcrossed **Figure 3—figure supplement 1F; Figures 3C, G and H; Figure 5A and C; Figure 6J-O** |
| Strain, strain background (*C. elegans*) | MT21793 | *Bhatla and Horvitz, 2015* | WBStrain00027622 | *lite-1(ce314) gur-3(ok2245)* **Figure 3—figure supplement 1G; Figure 3D and G and H** |
| Strain, strain background (*C. elegans*) | DA1113 | CGC | WBStrain00005547 | *eat-2(ad1113)* 2x outcrossed **Figure 3—figure supplement 1H and L** |
| Strain, strain background (*C. elegans*) | MT25982 | this study | n/a | *eat-2(ad1113); gur-3(ok2245)* **Figure 3—figure supplement 1H and L** |
| Strain, strain background (*C. elegans*) | MT22899 | this study | n/a | *eat-2(ad1113); lite-1(ce314)* **Figure 3—figure supplement 1I and L** |
| Strain, strain background (*C. elegans*) | MT25983 | this study | n/a | *eat-2(ad1113); lite-1(ce314) gur-3(ok2245)* **Figure 3—figure supplement 1J–1L** |
| Strain, strain background (*C. elegans*) | MT25999 | this study | n/a | *eat-2(ad1113); lite-1(ce314) gur-3(ok2245); nIs687 [lite-1$_{promoter}$::gfp::lite-1; lin-15(+)]* nIs687 was outcrossed 8x to MT22499 **Figure 3—figure supplement 1K and L** |
| Strain, strain background (*C. elegans*) | MT22499 | this study | n/a | *lite-1(ce314) gur-3(ok2245) lin-15AB(n765ts)* Used to outcross MT25999 |

*Continued on next page*

*Appendix 1—key resources table continued*

| Reagent type (species) or resource | Designation | Source or reference | Identifiers | Additional information |
|---|---|---|---|---|
| Strain, strain background (*C. elegans*) | MT25809 | this study | n/a | *lite-1(ce314) gur-3(ok2245) lin-15(n765ts); nEx2281 [lite-1$_{promoter}$::lite-1(+)::gfp; lin-15(+)]* **Figure 3E and Gand H** |
| Strain, strain background (*C. elegans*) | MT25810 | this study | n/a | *lite-1(ce314) gur-3(ok2245) lin-15(n765ts); nEx2157 [gur-3 gDNA; lin-15(+)]* **Figure 3F-H** |
| Strain, strain background (*C. elegans*) | MT23415 | this study | n/a | *nIs686 III; gur-3(ok2245) lin-15(n765ts)* **Figure 3I, J and M, N** |
| Strain, strain background (*C. elegans*) | MT23417 | this study | n/a | *nIs686 III; lite-1(ce314) lin-15(n765ts)* **Figure 3I, K and M, N** |
| Strain, strain background (*C. elegans*) | MT25631 | this study | n/a | *lin-15(n765ts); nIs864 [glr-2$_{promoter}$::gfp::unc-54 3' UTR; lin-15(+)]* 6x outcrossed **Figure 2—figure supplement 1D; Figure 4B and C; Figure 4—figure supplement 1E and F** |
| Strain, strain background (*C. elegans*) | MT25804 | this study | n/a | *oxIs322 [myo-2$_{promoter}$::mcherry::h2b; myo-3$_{promoter}$::mcherry::h2b] II; nIs310 [nlp-13$_{promoter}$::gfp] V; nEx2905* **Figure 4—figure supplement 1A and C** |
| Strain, strain background (*C. elegans*) | MT25950 | this study | n/a | *eat-2(ad1113); nIs865 (integrated transgene derived from nEx2905 [lury-1$_{promoter}$::ice::sl2::mcherry::unc-54 3' UTR; ttx-3$_{promoter}$::mcherry]; outcrossed 8x to N2)* **Figure 4—figure supplement 1D** |
| Strain, strain background (*C. elegans*) | MT25823 | this study | n/a | *lin-15AB(n765ts); nEx2917 [glr-2$_{promoter}$::hiscl::SL2::mcherry::unc-54 3' UTR; lin-15(+)]* **Figure 4H-L** |
| Strain, strain background (*C. elegans*) | MT23192 | this study | n/a | *lin-15(n765ts); nIs678 [glr-2$_{promoter}$::gcamp6s::unc-54 3' UTR; lin-15(+)]* 5x outcrossed. **Figure 4M–U, Figure 4—figure supplement 1W; Video 15** |
| Strain, strain background (*C. elegans*) | MT24110 | this study | n/a | *lin-15(n765ts); nIs780 [gur-3$_{promoter}$::gfp; lin-15(+)]* 7x outcrossed to MT8189 **Figure 3—figure supplement 1M and N** |
| Strain, strain background (*C. elegans*) | MT23250 | this study | n/a | *gur-3(ok2245) lin-15(n765ts); nIs678* **Figures 4P, Q, T and U** |

*Continued on next page*

*Appendix 1—key resources table continued*

| Reagent type (species) or resource | Designation | Source or reference | Identifiers | Additional information |
|---|---|---|---|---|
| Strain, strain background (*C. elegans*) | MT23230 | this study | n/a | *lite-1(ce314) lin-15(n765ts); nIs678* **Figures 4P, R, T and U**; **Figure 6D–I**; **Figure 6—figure supplement 1B–D** |
| Strain, strain background (*C. elegans*) | MT23369 | this study | n/a | *lite-1(ce314) gur-3(ok2245) lin-15(n765ts); nIs678* **Figures 4T and U**; **Figure 5D and E** |
| Strain, strain background (*C. elegans*) | MT23343 | this study | n/a | *unc-13(s69); lin-15(n765ts); nIs678* **Figure 4—figure supplement 1I and J** |
| Strain, strain background (*C. elegans*) | MT23410 | this study | n/a | *unc-31(u280); lin-15(n765ts); nIs678* **Figure 4—figure supplement 1K and L** |
| Strain, strain background (*C. elegans*) | MT25468 | this study | n/a | *lite-1(ce314) gur-3(ok2245) lin-15(n765ts); nEx2815 [lury-1$_{promoter}$::chr2::sl2:: mcherry::unc-54 3' UTR]* **Figure 4V and W** |
| Strain, strain background (*C. elegans*) | CB933 | Horvitz laboratory collection | WBStrain00004216 | *unc-17(e245)* **Figure 4—figure supplement 1M** |
| Strain, strain background (*C. elegans*) | PR1152 | Horvitz laboratory collection | WBStrain00030801 | *cha-1(p1152)* **Figure 4—figure supplement 1N** |
| Strain, strain background (*C. elegans*) | RM509 | CGC | WBStrain00033372 | *ric-3(md158)* **Figure 4—figure supplement 1O** |
| Strain, strain background (*C. elegans*) | CB156 | **Brenner, 1974**; **Takayanagi-Kiya and Jin, 2016** | WBStrain00004114 | *unc-25(e156) dnj-17(ju1162)* **Figure 4—figure supplement 1P** |
| Strain, strain background (*C. elegans*) | MT6308 | CGC | WBStrain00027259 | *eat-4(ky5)* **Figure 4—figure supplement 1Q** |
| Strain, strain background (*C. elegans*) | CB1112 | Horvitz laboratory collection | WBStrain00004246 | *cat-2(e1112)* **Figure 4—figure supplement 1R** |
| Strain, strain background (*C. elegans*) | MT10661 | Horvitz laboratory collection | WBStrain00027379 | *tdc-1(n3420)* **Figure 4—figure supplement 1S** |
| Strain, strain background (*C. elegans*) | MT14984 | Horvitz laboratory collection | WBStrain00027500 | *tph-1(n4622)* **Figure 4—figure supplement 1T** |
| Strain, strain background (*C. elegans*) | MT15434 | Horvitz laboratory collection | WBStrain00027519 | *tph-1(mg280)* **Figure 4—figure supplement 1U** |
| Strain, strain background (*C. elegans*) | MT22280 | Horvitz laboratory collection | MT22280 | *ser-4(ok512); mod-1(ok103); ser-1(ok345) ser-7(tm1325)* **Figure 4—figure supplement 1V** |

*Continued on next page*

*Appendix 1—key resources table continued*

| Reagent type (species) or resource | Designation | Source or reference | Identifiers | Additional information |
|---|---|---|---|---|
| Strain, strain background (*C. elegans*) | MT23606 | this study | n/a | *lite-1(ce314) gur-3(ok2245) lin-15AB(n765ts); nEx2144 [flp-15$_{promoter}$::mcherry::gur-3, lin-15(+)]* **Figure 5B and C** |
| Strain, strain background (*C. elegans*) | MT23545 | this study | n/a | *lite-1(ce314) gur-3(ok2245) lin-15AB(n765ts); nIs678; nEx2144* **Figure 5D-H** |
| Strain, strain background (*C. elegans*) | MT26001 | this study | n/a | *eat-2(ad1113); lite-1(ce314) gur-3(ok2245) nIs791 X (integrated transgene derived from [nEx2144: flp-15$_{promoter}$::mcherry::gur-3, lin-15(+)])* **Figure 5I** |
| Strain, strain background (*C. elegans*) | MT21421 | ***Bhatla and Horvitz, 2015*** | WBStrain00044110 | *nIs569 [flp-15$_{promoter}$::csp-1b; ges-1$_{promoter}$::gfp]* **Figure 6A** |
| Strain, strain background (*C. elegans*) | MT21791 | ***Bhatla and Horvitz, 2015*** | n/a | *lite-1(ce314) nIs569* **Figure 6B** |

