## [Decision Letter]

**Acceptance summary:**

A fundamental question in Neuroscience is how the nervous system controls different motor patterns for various behaviors. This work reports a surprising finding that distinct movements of the *C. elegans* pharynx, eating versus spitting, are controlled by compartmentalized calcium signaling in a single pharyngeal muscle. The authors further investigate the underlying neural circuits that control spitting versus eating behaviors.

**Decision letter after peer review:**

Thank you for submitting your article "An Hourglass Circuit Motif Transforms a Motor Program via Subcellularly Localized Calcium Signaling in Muscle" for consideration by *eLife*. Your article has been reviewed by 3 peer reviewers, one of whom is a member of our Board of Reviewing Editors, and the evaluation has been overseen by Piali Sengupta as the Senior Editor. The reviewers have opted to remain anonymous.

The reviewers have discussed the reviews with one another and the Reviewing Editor has drafted this decision to help you prepare a revised submission.

Summary:

Sando et al. extend on previous work by the same lab to delineate the neuronal mechanisms that control UV-light / ROS suppression of feeding and evoked spitting behaviors. They provide a nice characterization of pharyngeal behaviors that are involved in feeding and spitting, showing that upon UV-light stimulation feeding pumps are modulated to evoke spitting instead. M1 neurons are central to the spitting reflex; they sense light, integrate inputs from light sensitive I2 and I4 neurons and transmit the information to the anterior pharyngeal muscles pm1/2 and the anterior part of pm3. The conceptual advances of this paper are twofold: (1) the hourglass circuit motif as a means to transform ingestion movements into spits. (2) local activation of pm3 muscles via a compartmentalized calcium signal that ensures opening of only the anterior part of the alimentary tract. Most of the behavioral experiments are well done and the paper could be of potential interest to a broad audience. However, the reviewers raised some concerns that should be addressed prior to publication in *eLife*.

Essential revisions:

(1) A major concern is that all three reviewers are not convinced that the data presented here support the conclusion of local calcium dynamics in the anterior pm3 muscles. Since this is one of the major aspects of this study, it is essential to provide more experimental evidence. The authors used a pan-pharyngeal driver to express GCaMP. The imaging resolution seems not good enough to distinguish calcium transients in pm1/2/3 and the most straight forward interpretation of the results is that the anterior calcium transients are derived from pm1/2 but not pm3. It seems otherwise to rest on the claim that pm3 is sufficient for spitting and that, in the absence of pm1/2, local contraction of pm3 is the only way to hold the valve open during expulsion. Same for Figure 4F.

To substantiate the claim, these experiments should be repeated using a pm3 specific driver.

Alternatively, if pm3 specific drivers are not available, the experiments could be repeated upon laser ablation of pm1/2, to ensure that the signals are indeed specifically derived from pm3.

Perhaps, if imaging resolution and interference by emission light scattering permits, an overlay of a good DIC with GCaMP fluorescence may settle this more easily since pm3 stops at the base of the buccal cavity whereas pm1/2 line the cavity.

Individual recording traces of the different regions along with ethograms of the pharyngeal behaviors should be shown.

(2) The authors use a calcium imaging assay in immobilized worms to record UV-light evoked muscle activity- and pharyngeal neuron activity. While pumping and spitting behaviors occur at a frequency of up to 5Hz in the behavioral assays (e.g. Figure 1D,E), calcium dynamics in muscle and neurons were observed at 1-2 orders of magnitude slower (e.g. Figure 1 H,I; Figure 4H-M). However, the authors state that these dynamics would match well the time-scale at which light evoked pumps are observed. This is confusing. While it is possible that pharyngeal neurons encode the rate of pumping/spitting, muscle activity should correspond to the motor rhythms.

What is the pumping rate under the imaging/immobilization conditions? Do the animals spit? The behaviors under imaging conditions need to be better characterized and documented.

Individual traces should be shown throughout (like Figure 4H), importantly next to ethograms of pharyngeal behaviors.

The image acquisition rate should be stated in the methods? Was this also 2Hz like the flickering rate?

Only with this information at hand it is possible to properly interpret the imaging results. Are the measurements convoluted by low acquisition rate and slow on/off kinetics of GCaMP, or do light evoked pharyngeal behaviors occur at such a slow frequency in immobilized worms?

(3) The purported movements of the metastomal filter appear to be based solely on the observation of particle flow with a particular concentration and size of beads. At times this may be misleading. For example, the authors report that 25% of normal pumps are associated with openings of the metastomal filter. However, it is possible that the beads do not always become jammed in the buccal cavity, even if the metastomal flaps remain in position. Direct imaging of the metastomal flaps would address this question; if this is not possible the limitations of the assay should at least be acknowledged.

(4) The opening of the metastomal flaps during spitting is interpreted as a "rinsing" of its mouth "in response to a bad taste". This interpretation is problematic since the animal is "rinsing" its mouth with the same particles that have presumably induced the spitting. It would make more sense if the animal increased rather than decreased selectivity of the metastomal filter; this would allow water to enter the pharynx while excluding potentially toxic particles. If the authors insist in their interpretation they should at least discuss this issue.

(5) Line 183 – What is the basis for believing the sufficiency of pm3 is based on "contraction of a subcellular region"? And Line 188 – where is this "uncoupling" shown? There are few figures/data here. Is it deduced that this must be so because the pharyngeal valve is open while the lumen closes during spitting? Is local contraction of pm3 the only possible explanation for this? In the WT condition, for example, could pm1 and/or pm2 contraction overcome a global relaxation of pm3 to hold the valve upen during lumen closing? Although spitting apparently persists after ablation of pm1/pm2, these events should be documented in the same detail as WT events to demonstrate that pm3 is truly sufficient for "normal" spitting (i.e. continued pumping of lumen while the valve and filter are held open, local Ca++ events in anterior portion of pm3). This section seems to take a leap to a precise muscle mechanism based only on the ablation.

(6) At the cellular level, the authors note that calcium waves in muscle can cause local contraction patterns that lead to peristalsis, but that their observations seem to be of a different kind in terms of spatial and temporal patterning (long sustained local Ca++/contraction in one domain while rhythmic Ca/contraction occur in another domain). How input strength might create such a pattern is difficult to envision, given the simplicity of the M1 pm3 innervation pattern. What is the proposed cellular mechanism here?

(7) Figure 4J-L: these panels lack quantifications. Please show also individual traces; is the little initial bump in lite-1 mutants' response consistent across multiple recordings? Is the reduction in lite-1;gur-3 statistically significant?

Why is this initial transient signal so much stronger when gur-3 is expressed in I2 in the double mutants (Figure 5D)?

(8) Line 422-424: this statement is not supported by data in Figure 6B-F; only I4 ablated animals show a robust defect and there is no synergistic effect in the double ablation.

(9) Figure 6G: this result lacks quantifications. Appropriate statistics should be performed. Show also individual traces.

(10) Line 210 – "data not shown"….the correlation between spatially-restricted contraction / Ca++ signals and spitting is a central claim of the paper…it needs to be quantitatively documented in a figure.

(11) Line 104 – Is the experimenter blinded to strain / condition? If not, what steps were taken to detect or correct experimenter bias? This is a major pitfall of manual behavior coding.

---

## [Author Response]

Essential revisions:(1) A major concern is that all three reviewers are not convinced that the data presented here support the conclusion of local calcium dynamics in the anterior pm3 muscles. Since this is one of the major aspects of this study, it is essential to provide more experimental evidence. The authors used a pan-pharyngeal driver to express GCaMP. The imaging resolution seems not good enough to distinguish calcium transients in pm1/2/3 and the most straight forward interpretation of the results is that the anterior calcium transients are derived from pm1/2 but not pm3. It seems otherwise to rest on the claim that pm3 is sufficient for spitting and that, in the absence of pm1/2, local contraction of pm3 is the only way to hold the valve open during expulsion. Same for Figure 4F.To substantiate the claim, these experiments should be repeated using a pm3 specific driver.Alternatively, if pm3 specific drivers are not available, the experiments could be repeated upon laser ablation of pm1/2, to ensure that the signals are indeed specifically derived from pm3.Perhaps, if imaging resolution and interference by emission light scattering permits, an overlay of a good DIC with GCaMP fluorescence may settle this more easily since pm3 stops at the base of the buccal cavity whereas pm1/2 line the cavity.Individual recording traces of the different regions along with ethograms of the pharyngeal behaviors should be shown.

We have repeated the muscle calcium-imaging experiments using two different pm3-specific promoters in addition to the pan-pharyngeal muscle promoter we used previously. We observed compartmentalized calcium dynamics at the anterior end of the pm3 muscles in all three cases.

The *myo-2p* promoter expresses specifically in pm3 and in no other cell in the anterior pharynx (Okkema et al., 1993; Okkema and Fire, 1994). We obtained a strain that expresses GCaMP3 from the *myo-2p* promoter (Kozlova et al., 2019) and used confocal microscopy to confirm that these animals express GCaMP3 in pm3 but not in pm1 or pm2 (Figure 2—figure supplement 2B). After stimulation with light, 16/19 of the animals examined responded with subcellular calcium responses at the anterior tip of pm3 (11 of these 16 animals also showed a subsequent calcium increase in the posterior regions of the pharynx) (Figure 2K2L). These observations demonstrate that light induces subcellularly localized calcium increases in the anterior end of pm3 in response to light.

To confirm this conclusion, we sought a second pm3-specific promoter. We determined that the C*32F10.8p* promoter is expressed in the pm3s and the mc1 marginal cells, but not in pm1 or the pm2s. Because the pm3s and mc1s are similar in size and location, we combined calcium imaging with confocal microscopy of animals with mosaic *C32F10.8p::GCaMP3* expression to obtain videos in which we could confidently distinguish subcellular calcium signals in the pm3s from those in the mc1s. These experiments showed that the pm3s typically displayed subcellular responses to light (10/12 animals with an identifiable pm3 showed such a response), while the mc1s typically did not (only 1/7 animals with an identifiable mc1 showed such a response). We confirmed the GCaMP expression patterns for each strain using confocal microscopy (Figure 2—figure supplement 2A-2D). We have updated the text to describe these new findings (lines 277-338).

We have shown individual traces for all imaging experiments (i.e., Figures 2I, 2K, 2M, 3I, 4F, 4N, and 4P). Because our imaging system does not permit us to collect fluorescence and brightfield images simultaneously, we are unable to overlay imaging traces with ethograms of pharyngeal behavior (see discussion of the immobilized imaging assay under our response to Essential Revision 2 below).

(2) The authors use a calcium imaging assay in immobilized worms to record UV-light evoked muscle activity- and pharyngeal neuron activity. While pumping and spitting behaviors occur at a frequency of up to 5Hz in the behavioral assays (e.g. Figure 1D,E), calcium dynamics in muscle and neurons were observed at 1-2 orders of magnitude slower (e.g. Figure 1 H,I; Figure 4H-M). However, the authors state that these dynamics would match well the time-scale at which light evoked pumps are observed. This is confusing. While it is possible that pharyngeal neurons encode the rate of pumping/spitting, muscle activity should correspond to the motor rhythms.

We thank the reviewers for pointing out this confusion. During spitting, the contractions of those muscles near the pharyngeal valve (i.e., pm1, the pm2s, and the anterior ends of the pm3s) are much slower than the contractions of the muscles of the rest of the pharynx, i.e., these anterior muscles remain contracted as the rest of the pharynx rhythmically contracts and relaxes. The localized muscle calcium transients in the anterior pharynx that occur during spitting are similarly slow and match the time-scale of the muscle contractions that occur in that region. We have modified the text to clarify that the comparison is between the calcium data and contractions of the muscles adjacent to the pharyngeal valve (line 267-271).

What is the pumping rate under the imaging/immobilization conditions? Do the animals spit?

Animals do not pump spontaneously while immobilized, and we have modified the text to say so explicitly (lines 244-246). Light often triggers pumps by immobilized animals (Figure 2—figure supplement 1C and 1D), and these pumps result in the egestion of material held in the procorpus – i.e., these pumps are spits (Video 10).

While the light-induced pumping of immobilized animals carrying the pharyngeal muscle GCaMP transgene *nIs686* (*gpa-16p::GCaMP3)* was indistinguishable from that of the wild-type, the response of animals carrying the transgene *nEx3045* (*C32F10.8p::GCaMP3*) was significantly weaker, and animals carrying the transgene *cuIs36* (*myo-2p::GCaMP3*) did not pump at all when illuminated. The reduced responsivity in these two muscle GCaMP3-expressing strains we tested could be a consequence of GCaMP-mediated intracellular calcium-buffering or another abnormality induced by transgene overexpression.

We now reference these experiments in the Results (lines 246-251) and describe them in full in Materials and methods (lines 943-960) sections.

The behaviors under imaging conditions need to be better characterized and documented.Individual traces should be shown throughout (like Figure 4H), importantly next to ethograms of pharyngeal behaviors.

We have added individual traces throughout (Figure 2I, 2K, 2M, 3I, 4F, 4N, 4P, and Figure 6—figure supplement 1A). As noted above, we cannot obtain behavioral and calcium traces of the same animals, but we now include both multiple calcium traces (cited above) and multiple ethograms of light-induced pumping (Figure 2—figure supplement 1C).

The image acquisition rate should be stated in the methods? Was this also 2Hz like the flickering rate?

We have added the image acquisition rate (2 Hz) and exposure time (250 msec for all conditions with the exception of animals imaged in Figure 2—figure supplement 1F, which were imaged with an exposure time of 15 msec because of the very bright fluorescence produced by the *myo-2p::GFP* transgene these animals carried) to the Methods section (lines 961-966).

Only with this information at hand it is possible to properly interpret the imaging results. Are the measurements convoluted by low acquisition rate and slow on/off kinetics of GCaMP, or do light evoked pharyngeal behaviors occur at such a slow frequency in immobilized worms?

We now note in the Results (lines 264-267) and Discussion (lines 660-662) sections that our image acquisition rate (2 Hz) and exposure time (250 msec) are too slow to distinguish a sustained calcium increase from a series of short-lived calcium pulses. This ambiguity regarding the temporal pattern of calcium signaling does not change our interpretation of the spatial pattern of calcium signaling – during spitting behavior, calcium increases occur that are specifically localized to the anterior end of the pm3s.

(3) The purported movements of the metastomal filter appear to be based solely on the observation of particle flow with a particular concentration and size of beads. At times this may be misleading. For example, the authors report that 25% of normal pumps are associated with openings of the metastomal filter. However, it is possible that the beads do not always become jammed in the buccal cavity, even if the metastomal flaps remain in position. Direct imaging of the metastomal flaps would address this question; if this is not possible the limitations of the assay should at least be acknowledged.

We now explicitly state this limitation of the assay, i.e. that we inferred the open/closed state of the filter based on the movement of beads through the buccal cavity (lines 152-157). Our conclusions do not depend on the specific frequency with which the flaps open in feeding. Rather, we simply conclude that illumination changes the frequency of feeding pumps during which bead influx markedly increases, and we suggest that this increase occurs because of the opening of the metastomal filter.

(4) The opening of the metastomal flaps during spitting is interpreted as a "rinsing" of its mouth "in response to a bad taste". This interpretation is problematic since the animal is "rinsing" its mouth with the same particles that have presumably induced the spitting. It would make more sense if the animal increased rather than decreased selectivity of the metastomal filter; this would allow water to enter the pharynx while excluding potentially toxic particles. If the authors insist in their interpretation they should at least discuss this issue.

We agree that the ethology of a given behavior can be difficult to interpret from laboratory studies alone, and we now state explicitly that our “rinsing” hypothesis is speculative (lines 158-162 and 704-710). We also agree that the “rinsing” interpretation of filter-opening behavior might seem counterintuitive, and we have added clarification of our rationale for this interpretation (lines 707710). Specifically, both light and hydrogen peroxide (H_2_O_2_) not only induce spitting but also drive locomotory avoidance behaviors, which remove the worm from the site of the noxious stimulus (Edwards et al., 2008; Ward et al., 2008; Bhatla and Horvitz, 2015). Light and H_2_O_2_ rapidly inhibit pumping, and spitting occurs after a brief delay, so that most spits occur only as the animal is leaving or has already left the offending area. As noted by the reviewers, continued spitting/rinsing in a constant field of toxic material would likely be harmful to the animal. We have observed that spitting rate declines after about 5 sec of stimulation and stops completely after 10 sec. Perhaps this desensitization of the spitting reflex acts to prevent the animal from continuing to spit in the face of a sustained noxious stimulation.

(5) Line 183 – What is the basis for believing the sufficiency of pm3 is based on "contraction of a subcellular region"? And Line 188 – where is this "uncoupling" shown? There are few figures/data here. Is it deduced that this must be so because the pharyngeal valve is open while the lumen closes during spitting? Is local contraction of pm3 the only possible explanation for this? In the WT condition, for example, could pm1 and/or pm2 contraction overcome a global relaxation of pm3 to hold the valve upen during lumen closing? Although spitting apparently persists after ablation of pm1/pm2, these events should be documented in the same detail as WT events to demonstrate that pm3 is truly sufficient for "normal" spitting (i.e. continued pumping of lumen while the valve and filter are held open, local Ca++ events in anterior portion of pm3). This section seems to take a leap to a precise muscle mechanism based only on the ablation.

Our conclusion that there is an uncoupling of apparently separable subdomains of the pm3s during spitting is indeed an inference drawn from the premises that (1) spitting requires the prolonged opening of the anterior pharyngeal valve, (2) the pm3s are the only muscles positioned to hold this valve open when pm1 and the pm2s are ablated, and (3) the non-anterior regions of the pm3s rhythmically contract and relax during spitting. We now state this inference explicitly in the text (lines 198-205).

As the reviewers note, it is indeed possible that the contraction of pm1 and/or the pm2s is sufficient to open the pharyngeal valve in non-ablated animals. We now state this possibility in the text (lines 231-234).

We note that the ablation of pm1- and/or pm1/2 does have effects on pharyngeal behavior. Ablation of pm1 eliminates the metastomal filter, leading to greater ingestion of and increased amounts of material into the procorpus (lines 219-222; Figure 2—figure supplement 1A and 1B). pm1- and pm1/pm2-ablated animals often eject material less efficiently while spitting (lines 222-231). This inefficiency might stem from the overfilling of the pharynx because of loss of the filter (mock-ablated animals occasionally also fill their anterior pharynges and spit inefficiently when this is the case); from additional anatomical lesions in the anterior pharynx stemming from the ablation; and/or from a true functional requirement for pm1 (and, perhaps, pm2) in normal spitting. For these reasons, we cannot say whether pm3 suffices for truly wild-type spitting. Nonetheless, that pm1/2-ablated animals can open the pharyngeal valve does show that pm3 can suffice to drive spitting when those other muscles are absent.

We have added these considerations to the text (lines 206-236) and moved our observation of greater filling of the anterior pharynx in pm1-ablated animals to the main text (lines 219-231).

We also have added additional supplemental videos showing pm1- and pm1/2ablated animals spitting (Videos 7-9).

(6) At the cellular level, the authors note that calcium waves in muscle can cause local contraction patterns that lead to peristalsis, but that their observations seem to be of a different kind in terms of spatial and temporal patterning (long sustained local Ca++/contraction in one domain while rhythmic Ca/contraction occur in another domain). How input strength might create such a pattern is difficult to envision, given the simplicity of the M1 pm3 innervation pattern. What is the proposed cellular mechanism here?

We propose that the M1 neuromuscular junctions (NMJs) at the anterior end of the pm3 muscles drive the light-induced subcellular calcium increases, subcellular pm3 contraction, and subsequent global contractions of pharyngeal muscle. For M1 to evoke sustained subcellular contraction of the anterior ends of the pm3s while also stimulating pumps globally, this subcellular region must be functionally distinct from the posterior regions of the pm3s. For example, this anterior region might express a set of ion channels distinct from those expressed elsewhere in the pm3s or might be modulated by a locally-acting, M1-driven signaling cascade that inhibits the relaxation of the muscle segment at the anterior ends of the pm3s.

An alternative possibility is that M1 promotes subcellular contraction and global pumping via distinct sets of NMJs. In this model, M1’s NMJs with the anterior ends of the pm3s open the pharyngeal valve directly, while M1 indirectly stimulates global pumping by exciting another neuron that forms NMJs with a more posterior region of pharyngeal muscle. (The muscles of the pharynx are gap-junctioned together such that stimulation of one muscle could be sufficient to trigger a global contraction.) Eight pharyngeal neuron classes (I1, I2, I3, I5, M4, M5, MI, NSM), each of which forms NMJs with multiple pharyngeal muscle classes, receive synapses from M1 (Cook et al., 2020), and at least two of these classes (I1 and M4) are known to be able to promote pumping (Trojanowski et al., 2014) and thus could be plausible candidates to drive global, M1-dependent contractions in response to light. This second model does not require that the anterior tip of pm3 be functionally specialized beyond receiving NMJs from M1, because the threshold for local contraction could be lower than the threshold for depolarizing the entire muscle.

We have added these speculations to the Discussion (lines 664-689).

(7) Figure 4J-L: these panels lack quantifications. Please show also individual traces; is the little initial bump in lite-1 mutants' response consistent across multiple recordings? Is the reduction in lite-1;gur-3 statistically significant?

We have added quantifications and individual traces (Figure 4P, 4T, and 4U). The little initial bump in *lite-1* mutants is consistent across many recordings, and the reduction of this bump in *lite-1gur-3* double mutants as compared with *gur-3* single mutants is statistically significant (approximate adjusted P value <0.0001; Kruskal-Wallis test with Dunn’s multiple comparison test).

Why is this initial transient signal so much stronger when gur-3 is expressed in I2 in the double mutants (Figure 5D)?

Our guess is that M1 activation in the *lite-1 gur-3*; *I2p::gur-3* rescue strain appears to be greater than that in *lite-1* animals because M1 activation reflects the level of GUR-3 in the I2s and in the former case *gur-3* is overexpressed in the I2 neurons, whereas in the latter case *gur-3* is expressed by its endogenous promoter. We decided not to discuss this apparent difference because these observations were made on different days as parts of separate experiments, and this apparent difference is of no consequence to our conclusions. We note that the rescue strain overexpresses *gur-3* specifically in the I2 pharyngeal neurons and the PHA tail neurons (our imaging protocol illuminates only the head), while endogenous *gur-3* is expressed in several pharyngeal and head neurons (i.e., AVD, I4, M1, and MI) in addition to the I2s. Hence, an alternative hypothesis is that *gur-3* normally functions in a cell that directly or indirectly inhibits M1 or the I2s (such as MI or the PVDs), such that restricting *gur-3* function to the I2s (as the *pI2::gur-3* rescue strain does) eliminates this parallel *gur-3*-mediated inhibitory input to M1 and renders M1 more excitable.

We did not think this issue warrants the space it would take to discuss it, but would be happy to do so if that would be helpful, e.g., “We noted that the M1 calcium response in *lite-1gur-3*; *pI2::gur-3(+)* (Figure 5E) rescue animals appeared to be larger than that of *lite-1* single mutants (Figure 4R). This difference might reflect the known differences in the sites and levels of *gur-3* expression in these strains, but because these data were collected on different days we cannot confidently interpret this apparent difference.”

(8) Line 422-424: this statement is not supported by data in Figure 6B-F; only I4 ablated animals show a robust defect and there is no synergistic effect in the double ablation.

We regret the wording of this passage, which inadvertently asserted that the results shown in Figure 6B-F of the original submission (Figure 6D-6H of this revised submission) support a conclusion of synergy between the I2 and I4 neurons, which they do not. We have made the following corrections:

1. Our main observation from Figure 6D-H is that killing I4 reduces the spitting of *lite-1* I2-ablated animals, suggesting that I4 promotes spitting in *lite-1* I2ablated animals. This is the only conclusion we make from Figure 6H. I4 singly-ablated animals might be defective in spitting and enhanced in pumping inhibition relative to wild-type animals, but these differences are not statistically significant (Figure 6D, 6F, 6H, and 6I). Thus, our conclusion regarding I4 function in spitting is limited to a comparison between *lite-1* I2ablated and *lite-2* I2/I4 double-ablated animals. We also note two issues that generally affect the interpretation of neuronal ablations in *lite-1* animals with functional I2s, including I4 singly-ablated animals. First, when the I2 neurons are present they inhibit pumping, and this I2-dependent inhibition compresses the dynamic range of the assay by ~50% (Figure 6I). This compression has the potential to mask the contributions of non-I2 neurons to spitting and makes it difficult to accurately assay the effect of a given ablation on spitting. Second, when the I2s are present this assay cannot distinguish between an increase in inhibition and a decrease in spitting, even if statistically significant results are obtained. For these reasons, we do not wish to interpret any ablation in the context of functional I2s in a *lite-1* background. These limitations are specific to *lite-1* animals, because *lite-1* animals are so defective in pumping inhibition that the burst pumping phase is not clearly visible. We now state these limitations explicitly in the text (lines 564-574).

2. We removed all references to the qualitative timing of light-induced spitting. Specifically, we no longer describe the spitting of I2/I4-ablated animals as “delayed” in comparison to singly ablated animals and do not make any claims as to the effects of ablations on the timing of spitting.

3. Our reference to “redundant” function of the I2s in spitting on line 424 of our first submission was intended to reference both Figure 5, which shows that the I2 neurons promote spitting via M1 (Figure 5A-5G) and Figure 6, which suggests that I4 is required for spitting in I2-ablated animals (Figure 6E, 6G, and 6H). In other words, we intended this statement to be an inference drawn from multiple experiments and not simply from Figure 6D-6H. We removed this problematic statement. A better word than “redundant” would have been “parallel,” and we have made this change to the concluding sentence (lines 598-600) of the subsection (“In short, the I2 and I4 neurons and perhaps others likely function in parallel to promote *gur-3*-, M1-dependent spitting”). We have also added quantification to show this effect of the I2s on global pumping rate in this experiment (Figure 6I) and have modified or removed other language in this section to improve clarity (lines 554-574).

(9) Figure 6G: this result lacks quantifications. Appropriate statistics should be performed. Show also individual traces.

We did not detect statistically significant differences among the four experimental conditions in question. We now note this lack of significance (lines 575-579) and have moved these data (now with quantification and individual traces) to a figure supplement (Figure 6—figure supplement 1A-1C).

(10) Line 210 – "data not shown"….the correlation between spatially-restricted contraction / Ca++ signals and spitting is a central claim of the paper…it needs to be quantitatively documented in a figure.

We now document this result with a supplemental video (Video 12) and figure panel (Figure 2—figure supplement 1E).

(11) Line 104 – Is the experimenter blinded to strain / condition? If not, what steps were taken to detect or correct experimenter bias? This is a major pitfall of manual behavior coding.

The manually-coded experiments (main Figures 1B and 5I and figure supplements 3—1A-1L, 4—1B, 1D, 1M-1V, and 4—2A-2P’) were not performed blinded. As part of our prior efforts to assess and guard against experimenter biases, we previously demonstrated that even subtle pumping phenotypes can be recorded by eye with a fidelity similar to that obtained by video camera (Bhatla and Horvitz, 2015 – see Figure S1A-C). In our current study, we performed many experiments using both real-time manual scoring and video capture. In all cases but one, the results of the two methods were equivalent (i.e., Figure 1B is reproduced by Figure 1D and 1E and figure supplements 3—1C, 1F, and 1G are reproduced by Figure 3A-3D). In one case, there was a minor difference, but this difference involved light-induced pumping inhibition behavior and not burst pumping (cf. Figure 4—figure supplement 1B and 1G). Because the spitting and burst pumping assays differ in several ways (i.e., the spitting assay involves the selective illumination of the heads of individual worms feeding in the presence of polystyrene beads, while the burst pumping assay illuminates the entire bodies of animals feeding on a bacterial lawn), we cannot definitively attribute this difference to scoring method. Furthermore, both experiments supported the same major conclusion – genetic ablation of M1 eliminates burst pumping.

In some experiments, behavior was manually coded in real-time without the collection of corresponding videos. These experiments were of two categories: some involved the interpretation of large all-or-nothing differences highly likely to be robust to experimenter error and/or bias (i.e., Figure 5I; Figure 3—figure supplement 1H-1L, Figure 4—figure supplement 1M-1O), while the remaining cases (Figure 3—figure supplement 1A, 1B, 1D, and 1E; Figure 4—figure supplement 1P-1V; and Figure 4—figure supplement 2A-2P’) were peripheral to the primary findings of this study and typically reported negative results.

We have added text explicitly acknowledging these limitations to the Methods section, along with a list of experiments scored by eye followed by a list of experiments scored by both eye and video (lines 863-870).